# ATTENTIONAL CONSTELLATION NETS FOR FEW-SHOT LEARNING

**Weijian Xu**[*,1], **Yifan Xu**[*,1], **Huaijin Wang**[*,1] **& Zhuowen Tu**[1,2]
University of California San Diego[1], Amazon Web Services[2]
`{wex041,yix081,huw011,ztu}@ucsd.edu`

## ABSTRACT

The success of deep convolutional neural networks builds on top of the learning of effective convolution operations, capturing a hierarchy of structured features via filtering, activation, and pooling. However, the explicit structured features, e.g. object parts, are not expressive in the existing CNN frameworks. In this paper, we tackle the few-shot learning problem and make an effort to enhance structured features by expanding CNNs with a constellation model, which performs cell feature clustering and encoding with a dense part representation; the relationships among the cell features are further modeled by an attention mechanism. With the additional constellation branch to increase the awareness of object parts, our method is able to attain the advantages of the CNNs while making the overall internal representations more robust in the few-shot learning setting. Our approach attains a significant improvement over the existing methods in few-shot learning on the CIFAR-FS, FC100, and *mini*-ImageNet benchmarks.

## 1 INTRODUCTION

Tremendous progress has been made in both the development and the applications of the deep convolutional neural networks (CNNs) (Krizhevsky et al., 2012; Simonyan & Zisserman, 2015; Szegedy et al., 2015; He et al., 2016; Xie et al., 2017). Visualization of the internal CNN structure trained on e.g. ImageNet (Deng et al., 2009) has revealed the increasing level of semantic relevance for the learned convolution kernels/filters to the semantics of the object classes, displaying bar/edge like patterns in the early layers, object parts in the middle layers, and face/object like patterns in the higher layers (Zeiler & Fergus, 2014). In general, we consider the learned convolution kernels being somewhat implicit about the underlying objects since they represent projections/mappings for the input but without the explicit knowledge about the parts in terms of their numbers, distributions, and spatial configurations.

On the other hand, there has been a rich history about explicit object representations starting from deformable templates (Yuille et al., 1992), pictorial structure (Felzenszwalb & Huttenlocher, 2005), constellation models (Weber et al., 2000; Fergus et al., 2003; Sudderth et al., 2005; Fei-Fei et al., 2006), and grammar-based model (Zhu & Mumford, 2007). These part-based models (Weber et al., 2000; Felzenszwalb & Huttenlocher, 2005; Fergus et al., 2003; Sudderth et al., 2005; Zhu & Mumford, 2007) share three common properties in the algorithm design: (1) *unsupervised learning*, (2) *explicit clustering* to obtain the *parts*, and (3) modeling to characterize the *spatial configuration of the parts*. Compared to the CNN architectures, these methods are expressive with explicit part-based representation. They have pointed to a promising direction for object recognition, albeit a lack of strong practice performance on the modern datasets. Another line of object recognition system with the part concept but trained discriminatively includes the discriminative trained part-based model (DPM) (Felzenszwalb et al., 2009) and the spatial pyramid matching method (SPM) (Lazebnik et al., 2006). In the context of deep learning, efforts exist to bring the explicit part representation into deep hierarchical structures (Salakhutdinov et al., 2012).

The implicit and explicit feature representations could share mutual benefits, especially in *few-shot learning* where training data is scarce: CNNs may face difficulty in learning a generalized representation due to lack of sufficient training data, whereas clustering and dictionary learning

---

*indicates equal contribution

provide a direct means for data abstraction. In general, end-to-end learning of both the implicit and explicit part-based representations is a viable and valuable means in machine learning. We view convolutional features as an implicit part-based representation since they are learned through back-propagation via filtering processes. On the other hand, an explicit representation can be attained by introducing feature clustering that captures the data abstraction/distribution under a mixture model.

In this paper, we develop an end-to-end framework to combine the implicit and explicit part-based representations for the few-shot classification task by seamlessly integrating constellation models with convolution operations. In addition to keeping a standard CNN architecture, we also employ a cell feature clustering module to encode the potential object parts. This procedure is similar to the clustering/codebook learning for appearance in the constellation model (Weber et al., 2000). The cell feature clustering process generates a dense distance map. We further model the relations for the cells using a self-attention mechanism, resembling the spatial configuration design in the constellation model (Weber et al., 2000). Thus, we name our method constellation networks (ConstellationNet). We demonstrate the effectiveness of our approach on standard few-shot benchmarks, including FC100 (Oreshkin et al., 2018), CIFAR-FS (Bertinetto et al., 2018) and *mini*-ImageNet (Vinyals et al., 2016) by showing a significant improvement over the existing methods. An ablation study also demonstrates the effectiveness of ConstellationNet is not achieved by simply increasing the model complexity using e.g. more convolution channels or deeper and wider convolution layers (WRN-28-10 (Zagoruyko & Komodakis, 2016)) (see ablation study in Table 3 and Figure 2 (e)).

## 2 RELATED WORK

**Few-Shot Learning.** Recently, few-shot learning attracts much attention in the deep learning community (Snell et al., 2017; Lee et al., 2019). Current few-shot learning is typically formulated as a *meta-learning* problem (Finn et al., 2017), in which an effective feature embedding is learned for generalization across novel tasks. We broadly divide the existing few-shot learning approaches into three categories: (1) *Gradient-based methods* optimize feature embedding with gradient descent during meta-test stage (Finn et al., 2017; Bertinetto et al., 2018; Lee et al., 2019). (2) *Metric-based methods* learn a fixed optimal embedding with a distance-based prediction rule (Vinyals et al., 2016; Snell et al., 2017). (3) *Model-based methods* obtains a conditional feature embedding via a weight predictor (Mishra et al., 2017; Munkhdalai et al., 2017). Here we adopt ProtoNet (Snell et al., 2017), a popular metric-based framework, in our approach and boost the generalization ability of the feature embeddings with explicit structured representations from the constellation model. Recently, Tokmakov et al. (2019) proposes a compositional regularization to the image with its attribute annotations, which is different from out unsupervised part-discovery strategy.

**Part-Based Constellation/Discriminative Models.** The constellation model family (Weber et al., 2000; Felzenszwalb & Huttenlocher, 2005; Fergus et al., 2003; Sudderth et al., 2005; Fei-Fei et al., 2006; Zhu & Mumford, 2007) is mostly generative/expressive that shares two commonalities in the representation: (1) clustering/codebook learning in the appearance and (2) modeling of the spatial configurations. The key difference among these approaches lies in how the spatial configuration is modeled: Gaussian distributions (Weber et al., 2000); pictorial structure (Felzenszwalb & Huttenlocher, 2005); joint shape model (Fergus et al., 2003) ; hierarchical graphical model (Sudderth et al., 2005); grammar-based (Zhu & Mumford, 2007). These constellation models represent a promising direction for object recognition but are not practical competitive compared with deep learning based approaches. There are also discriminative models: The discriminatively trained part-based model (DPM) (Felzenszwalb et al., 2009) is a typical method in this vein where object parts (as HOG features (Dalal & Triggs, 2005)) and their configurations (a star model) are learned jointly in a discriminative way. The spatial pyramid matching method (SPM) (Lazebnik et al., 2006) has no explicit parts but instead builds on top of different levels of grids with codebook learned on top of the SIFT features (Lowe, 2004). DPM and SPM are of practical significance for object detection and recognition. In our approach, we implement the constellation model with cell feature clustering and attention-based cell relation modeling to demonstrate the appearance learning and spatial configuration respectively.

Parts models are extensively studied in fine-grained image classifications and object detection to provide spatial guidance for filtering uninformative object proposals (Simon & Rodner, 2015; Peng et al., 2017; Zhu et al., 2017; Ge et al., 2019; Qi et al., 2019). Related to our work, Neural Activation Constellations (NAC) (Simon & Rodner, 2015) introduces the constellation model to perform unsupervised part model discovery with convolutional networks. Our work is different from NAC in three aspects: (1) The algorithmic mechanisms behind Simon & Rodner (2015) and ours are

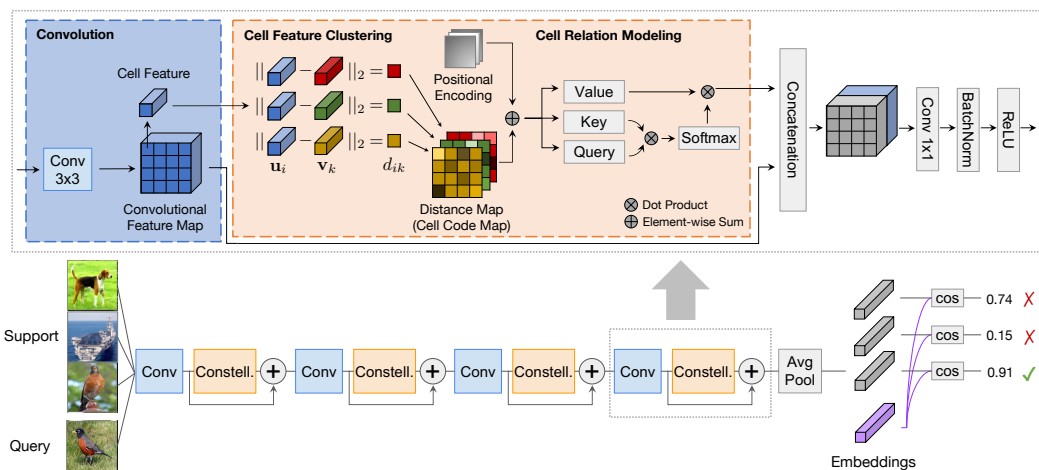

Figure 1: **Illustration of our ConstellationNet pipeline** where the bottom part is the network architecture based on Conv-4 backbone, and the top part shows the constellation model. Our proposed ConstellationNet consists of "Constell." modules that perform explicit cell feature clustering with self-attention for joint relation modeling.

different. Simon & Rodner (2015) implements a traditional Gaussian-based constellation module to model the spatial configuration and part selection on top of a fixed pre-trained CNN. However, in our ConstellationNet, our part representation and spatial configuration are modeled by cell feature clustering and self-attention based cell relation module, which is general-purpose, modularized and recursive. (2) In Simon & Rodner (2015) , the constellation module is optimized in an EM-like algorithm, which is separate from the CNN optimization. Our constellation modules are seamlessly integrated into the current CNNs and jointly optimized with them. (3) Our ConstellationNet uses the dense cell features from the CNN feature maps, which considers all positions from the images as potential parts and models their relation. However, (Simon et al. 2015) extracts sparse part representations (i.e. it uses at most one part proposal per channel and selects even less parts later), which may not fully utilize the rich information from the CNN feature maps.

## 3 FEW-SHOT LEARNING

In a standard classification problem, we aim to learn a model trained on the dataset $\mathcal{D}^{\text{base}}$ that can generalize its classification ability to unseen test set $\mathcal{D}^{\text{novel}}$ belonging to same categories. In few-shot classification problem, we encourage $\mathcal{D}^{\text{base}}$ and $\mathcal{D}^{\text{novel}}$ to be formed from different categories to emphasize model's generalization ability on novel categories, where we denote training categories as $\mathcal{C}_{\text{base}}$, test categories as $\mathcal{C}_{\text{novel}}$, and $\mathcal{C}_{\text{base}} \cap \mathcal{C}_{\text{novel}} = \varnothing$ to ensure the fairness.

In the training stage (a.k.a. *meta-train* stage), metric-based few-shot learning approaches (Snell et al., 2017; Vinyals et al., 2016; Oreshkin et al., 2018) usually learn a feature extractor $\phi(\mathbf{x})$ on the dataset $\mathcal{D}^{\text{base}}$ to obtain generic feature embedding by optimizing the loss $\mathcal{L}(\phi)$:

$$\mathcal{L}(\phi) = \mathbb{E}_{\{(\mathbf{x},y)\} \sim \mathcal{D}_{\text{base}}} \ell\big(\{(\phi(\mathbf{x}), y)\}\big) \tag{1}$$

where $\{(\mathbf{x}, y)\}$ is a sampled mini-batch of data points and $\ell(\cdot)$ is usually an episodic few-shot loss (Vinyals et al., 2016) or a standard cross-entropy loss (Chen et al., 2020).

In the inference stage (a.k.a. *meta-test* stage), a typical few-shot benchmark evaluates the model on $K$-way, $N$-shot classification tasks $\mathcal{T}$ drawn from $\mathcal{D}^{\text{novel}}$, where each task has a support set and a query set, i.e. $\mathcal{T} = (\mathcal{T}^{\text{supp}}, \mathcal{T}^{\text{query}})$. The support set $\mathcal{T}^{\text{supp}}$ contains $K$ classes and each class has $N$ images (e.g. $K = 5$, $N \in \{1, 5\}$). Following Snell et al. (2017), the prediction $\hat{y}'$ of a query image $\mathbf{x}' \in \mathcal{T}^{\text{query}}$ is given by the label of nearest prototype $\mathbf{c}_k$ from $\mathcal{T}^{\text{supp}}$ under a cosine similarity $d(\cdot, \cdot)$:

$$\hat{y}' = \arg\max_k d\big(\phi(\mathbf{x}'), \mathbf{c}_k\big), \qquad \mathbf{c}_k = \frac{1}{N} \sum_{(\mathbf{x},y) \in \mathcal{T}^{\text{supp}}, \ y=k} \phi(\mathbf{x}). \tag{2}$$

An extended description of the few-shot learning framework can be found from Appendix A.1. The generalization ability of the feature extractor $\phi(\mathbf{x})$ is improved in terms of training scheme (e.g.

episodic learning (Vinyals et al., 2016)), network design (e.g. task condition (Oreshkin et al., 2018)) or objective function (e.g. learnable distance (Sung et al., 2018)). In our method, we propose a novel network design by inserting constellation models into CNNs and strengthen the intermediate features.

## 4 CONSTELLATION MODEL

The concept of constellation has been introduced to the few-shot learning scenario in early years (Fei-Fei et al., 2006), in which the appearance and the shape are independently learned in a mixture model. In our work, we revisit the constellation model in an end-to-end learning framework: First, we define the a *cell feature* as the individual local feature at a position in the feature map (see Figure 1). We then employ *cell feature clustering* to model the underlying distribution of input cell features, implying a part discovery procedure. We further obtain the distance map of the cell features from clustering and then perform *cell relation modeling* to build spatial relationships.

### 4.1 CELL FEATURE CLUSTERING

In convolutional neural networks (CNNs), the convolutional filters are learned to detect the discriminative patterns from low-level to high-level through back-propagation (Zeiler & Fergus, 2014). In fact, the backward signal in the back-propagation is not necessarily needed to obtain a pattern detector. With the feature map in the forward step of the CNN, we are able to cluster the individual features at each location of the feature map (a.k.a. *cell features*) into multiple centers and employ the cluster centers as filters (Coates & Ng, 2012; Krähenbühl et al., 2015). Assume we obtain a convolutional feature map $\mathbf{U}$ with batch size $B$, spatial size $H \times W$ and channels $C$. We disensemble the feature map $\mathbf{U} \in \mathbb{R}^{B \times H \times W \times C}$ into a *cell features set* $\mathcal{U} = \{\mathbf{u}_1, \mathbf{u}_2, ..., \mathbf{u}_n\}$ where $n = BHW$ and $\mathbf{u}_i \in \mathbb{R}^C$ is a cell feature. Naively, we can conduct a $k$-means algorithm on input cell features $\mathcal{U}$ to solve the clustering objective:

$$\min \sum_i \sum_k m_{ik} ||\mathbf{u}_i - \mathbf{v}_k||_2^2 \quad \text{s.t.} \quad m_{ik} \in \{0,1\}, \quad \sum_k m_{ik} = 1 \tag{3}$$

where $\mathcal{V} = \{\mathbf{v}_1, \mathbf{v}_2, ..., \mathbf{v}_K\}$ is a set of cluster centers and $m_{ik}$ indicates if the input cell feature $\mathbf{u}_i$ is assigned to cluster center $\mathbf{v}_k$. The clustering-based filters $\mathcal{V}$ can model the underlying cell feature distributions and capture the most frequent features, which can be explicitly interpreted as meaningful part patterns/part types. The hard assignment map $\mathbf{m}_i = (m_{i1}, m_{i2}, ..., m_{iK})$ of input cell feature $\mathbf{u}_i$ onto the cluster centers can be used as a part-based representation, providing alternative information to the next layer in the CNN.

However, there are two issues remaining unsolved in the naive design: Firstly, CNNs are typically optimized in a stochastic gradient descent (SGD) manner. Thus, in each forward step, only a mini-batch of images are proceeded to provide cell features, which implies that the cluster centers cannot extract the global feature distribution across the whole dataset. Secondly, the hard assignment map has limited information due to its discrete representation. Therefore, inspired by Sculley (2010), we design a mini-batch soft $k$-means algorithm to cluster the cell features approximately:

- **Initialization.** Randomly initialize global cluster centers $\mathcal{V} = \{\mathbf{v}_1, \mathbf{v}_2, ..., \mathbf{v}_K\}$ and a counter $\mathbf{s} = (s_1, s_2, ..., s_K) = \mathbf{0}$.

- **Cluster Assignment.** In forward step, given input cell features $\mathcal{U} = \{\mathbf{u}_1, \mathbf{u}_2, ..., \mathbf{u}_n\}$, we compute the distance vector $\mathbf{d}_i = (d_{i1}, d_{i2}, ...d_{iK})$ between input cell feature $\mathbf{u}_i$ and all cluster centers $\mathcal{V}$. We then compute the soft assignment $m_{ik} \in \mathbb{R}$ and generate the current mini-batch centers $\mathbf{v}'_k$:

$$d_{ik} = ||\mathbf{u}_i - \mathbf{v}_k||_2^2, \qquad m_{ik} = \frac{e^{-\beta d_{ik}}}{\sum_j e^{-\beta d_{ij}}}, \qquad \mathbf{v}'_k = \frac{\sum_i m_{ik}\mathbf{u}_i}{\sum_i m_{ik}} \tag{4}$$

  where $\beta > 0$ is an inverse temperature.

- **Centroid Movement.** We formulate a count update $\Delta \mathbf{s} = \sum_i \mathbf{m}_i$ by summing all assignment maps $\mathbf{m}_i = (m_{i1}, m_{i2}, ...m_{iK})$. The current mini-batch centers $\mathbf{v}'_k$ are then updated to the global centers $\mathbf{v}_k$ with a momentum coefficient $\eta$:

$$\mathbf{v}_k \leftarrow (1 - \eta)\mathbf{v}_k + \eta\mathbf{v}'_k, \qquad \eta = \frac{\lambda}{s_k + \Delta s_k} \tag{5}$$

- **Counter Update.** Counter $\mathbf{s}$ is updated and distance vectors $\{\mathbf{d}_i\}$ are reshaped and returned:

$$\mathbf{s} \leftarrow \mathbf{s} + \Delta \mathbf{s} \tag{6}$$

With gradually updating global cluster centers, the above algorithm is able to address the issue of limited data in a mini-batch. In addition, we reshape the distance vectors $\{\mathbf{d}_i\}$ of all input cell features to a distance map $\mathbf{D} \in \mathbb{R}^{B \times H \times W \times K}$. Each distance vector $\mathbf{d}_i$ can be seen as a *learned cell code* in codebook (dictionary) learning, which encodes a soft assignment of the visual word (i.e. cell feature) onto the codewords (i.e. cluster centers) and implies a part representation. The distance map $\mathbf{D}$ then can be viewed as a cell code map that represents a spatial distribution of identified parts, which is passed to following layers. Empirically, it is observed that when $\mathbf{u}_i$ and $\mathbf{v}_k$ are $L_2$ normalized, the training procedure is more stable and the Euclidean distance $d_{ik}$ is equivalent to a cosine similarity up to an affine transformation. Details of the cell feature clustering can be found in Appendix A.9.

## 4.2 CELL RELATION AND SPATIAL CONFIGURATION MODELING

Before the deep learning era, traditional constellation models (Fei-Fei et al., 2006) decompose visual information into appearance and shape representation. The appearance of different parts in the image is treated independently while the shape of parts is assumed to have spatial connections. In our constellation model, we establish the spatial relationship among the individual part-based representations at a different location from the distance map as well. Specifically, we apply the self-attention mechanism (Vaswani et al., 2017) to build the spatial relationship and enhance the representation instead of using probabilistic graphical models in prior work (Fei-Fei et al., 2006).

In cell relation modeling, we add a *positional encoding* $\mathbf{P} \in \mathbb{R}^{B \times H \times W \times C}$ following Carion et al. (2020) for spatial locations to the distance map $\mathbf{D}$ and obtain the input feature map $\mathbf{F}_{\mathrm{I}}$ for query and key layers. For value layer, we directly flatten the distance map $\mathbf{D}$ to another input feature map $\mathbf{F}'_{\mathrm{I}}$:

$$\mathbf{F}_{\mathrm{I}} = \mathrm{SpatialFlatten}(\mathbf{D} + \mathbf{P}) \in \mathbb{R}^{B \times HW \times K}, \quad \mathbf{F}'_{\mathrm{I}} = \mathrm{SpatialFlatten}(\mathbf{D}) \in \mathbb{R}^{B \times HW \times K} \quad (7)$$

The input feature maps $\mathbf{F}_{\mathrm{I}}, \mathbf{F}'_{\mathrm{I}}$ are transformed into query, key and value $\{\mathbf{F}^q, \mathbf{F}^k, \mathbf{F}^v\} \subset \mathbb{R}^{B \times HW \times K}$ by three linear layers $\{\mathbf{W}^q, \mathbf{W}^k, \mathbf{W}^v\} \subset \mathbb{R}^{K \times K}$ and further computes the output feature $\mathbf{F}_{\mathrm{A}}$:

$$[\mathbf{F}^q, \mathbf{F}^k, \mathbf{F}^v] = [\mathbf{F}_{\mathrm{I}}\mathbf{W}^q, \mathbf{F}_{\mathrm{I}}\mathbf{W}^k, \mathbf{F}'_{\mathrm{I}}\mathbf{W}^v] \quad (8)$$

$$\mathbf{F}_{\mathrm{A}} = \mathrm{Att}(\mathbf{F}^q, \mathbf{F}^k, \mathbf{F}^v) = \mathrm{softmax}\Big(\frac{\mathbf{F}^q(\mathbf{F}^k)^\top}{\sqrt{K}}\Big)\mathbf{F}^v \quad (9)$$

The softmax of dot product between query and key matrix $\mathbf{F}^q(\mathbf{F}^k)^\top \in \mathbb{R}^{B \times HW \times HW}$ calculates the similarity scores in the embedding space among features across the spatial dimension. This encodes the spatial relationships of input features and leads to an enhanced output feature representation $\mathbf{F}_{\mathrm{A}}$. Besides, $\sqrt{K}$ in the denominator is to stabilize the gradient. In practice, we adopt a multi-head attention to model the feature relation in the embedding subspaces:

$$\mathbf{F}_{\mathrm{MHA}} = \mathrm{MultiHeadAtt}(\mathbf{F}^q, \mathbf{F}^k, \mathbf{F}^v) = [\mathbf{F}_1, ..., \mathbf{F}_J]\mathbf{W}, \qquad \mathbf{F}_j = \mathrm{Att}(\mathbf{F}^q_j, \mathbf{F}^k_j, \mathbf{F}^v_j) \quad (10)$$

In a $J$-head attention, the aforementioned similarity scores in the $K' = \frac{K}{J}$ dimensional embedding subspace are calculated using the query, key and value from $j$-th head, i.e. $\{\mathbf{F}^q_j, \mathbf{F}^k_j, \mathbf{F}^v_j\} \subset \mathbb{R}^{B \times HW \times K'}$. The output features $\mathbf{F}_j$ of each head are computed following Eq. 9. All the output features $\{\mathbf{F}_1, ..., \mathbf{F}_J\}$ are concatenated back into $K$ dimension embedding and further processed with a linear layer $\mathbf{W} \in \mathbb{R}^{K \times K}$ to generate multi-head output features $\mathbf{F}_{\mathrm{MHA}}$. Such multi-head attention settings could provide more diverse feature relation without introducing extra parameters.

## 4.3 INTEGRATE CONSTELLATION MODEL WITH CNNS

Our constellation model has the capability to capture explicit structured features and encodes spatial relations among the cell features. The output features yield informative visual cues which are able to strengthen the convolutional features. Thus, as shown in Figure 1, we place the constellation model after the convolution operation to extract its unique explicit features and concatenate them with the original convolutional feature map. A following $1 \times 1$ convolutional layer is used on the concatenated features to restore the channels of convolutional feature map. In Table 3, we provide evidence that merging features from constellation model to the CNN backbone can significantly improve the representation ability. In contrast, increasing channels in CNNs alone to double the parameters (second row in Table 3) can only improve the performance marginally. Optionally, we found it is useful to adopt auxiliary loss when training the constellation model in deeper networks (e.g. ResNet-12). On top of each constellation model, we conduct a standard classification to acquire additional regularization.

### 4.4 WHY CLUSTERING AND SELF-ATTENTION (CLUSTERING MAP + POSITIONAL ENCODING)?

As described in Section 1 and 2, classical constellation models (Fergus et al., 2003; Felzenszwalb & Huttenlocher, 2005) extract parts with their spatial relationships; they are expressive but do not produce competitive results on modern image benchmarks. CNN models (Krizhevsky et al., 2012; He et al., 2016) attain remarkable results on large-scale image benchmarks (Deng et al., 2009) but they are limited when training data is scarce. We take the inspiration from the traditional constellation models, but with a realization that overcomes their previous modeling limitations.

The main contribution of our work is a constellation module/block that performs **cell-wise clustering**, followed by **self-attention** on the **clustering distance map + positional encoding**. This separates our work from previous attempts, e.g. non-local block work (Wang et al., 2018) in which long-range non-linear averaging is performed on the convolution features (no clustering, nor positional encoding for the spatial configuration). The main properties of our constellation block include: (1) **Cell based dense representation** as opposed to the sparse part representation in (Weber et al., 2000) to make the cells recursively modeled in the self-attention unit in a modularized and general-purpose way. (2) **Clustering** to generate the cell code after clustering (codebook learning) that attains abstraction and is not dependent on the CNN feature dimensions. (3) **Positional encoding** (as in Carion et al. (2020)) for cells to encode the spatial locations. (4) **Tokenized representation** as expressive parts (code/clustering distance map + positional encoding) for the cells. (5) **Self-attention** to jointly model the cell code and positional encoding to capture the relationships between the parts together with their spatial configurations.

## 5 EXPERIMENT

### 5.1 DATASETS

We adopt three standard benchmark datasets that are widely used in few-shot learning, CIFAR-FS dataset (Bertinetto et al., 2018), FC100 dataset (Oreshkin et al., 2018), and *mini*-ImageNet dataset (Vinyals et al., 2016). Details about dataset settings in few-shot learning are in Appendix A.2.

### 5.2 NETWORK WITH MULTI-BRANCH

We build ConstellationNet on two ProtoNet variants, namely Conv-4 and ResNet-12, which are commonly used in few-shot learning. Details of networks and the optimization are in Appendix.

We develop a new technique, *Multi-Branch*, to optimize standard classification loss and prototypical loss simultaneously. We find the two training schemes, standard classification scheme and prototypical scheme, can be a companion rather than a conflict. Details of these two schemes can be found from Appendix A.1. Different from standard network backbone used in prior works, our embedding $\phi(\mathbf{x})$ is separated into two branches after a shared stem (Y-shape). Details of our multi-branch design are elaborated in A.10. The detailed ablation study is described in Table 3.

*Feature Augmentation.* During the meta-testing stage, we discover that concatenating features before average pooling to the final output can improve classification accuracy. The advantage of this technique is that no additional training and model parameters are introduced.

### 5.3 RESULTS ON STANDARD BENCHMARKS

Table 1 and 2 summarize the results of the few-shot classification tasks on CIFAR-FS, FC100, and *mini*-ImageNet, respectively. Our method shows a notable improvement over several strong baselines in various settings. ConstellationNet significantly improves the performance on shallow networks (Conv-4). In Table 2, our model outperforms SIB (Hu et al., 2020) 1-shot by 0.6% and 5-shot by 5.6%. In Table 1, our model outperforms MetaOptNet (Lee et al., 2019) by 5.95% in 1-shot and 6.24% in 5-shot. For deep networks with rich features, the constellation module still contributes to the performance, showing its complementary advantage to convolution. Our ResNet-12 model beats (Lee et al., 2019) 1-shot result by 2.7% on FC100, 3.4% on CIFAR-FS, and 1.72% on *mini*-ImageNet. The consistent improvement over both shallow and deep networks across all three datasets shows the generality of our method. Our ConstellationNet is orthogonal to the margin loss based methods (Liu et al., 2020; Li et al., 2020), and we also do not use extra cross-modal information (Xing et al., 2019; Li et al., 2020). On the contrary, our model enhances the embedding generalization ability by incorporating its own part-based representation. Additionally, to verify the orthogonality of our method, we adapt the negative margin loss following Liu et al. (2020) to our Conv-4 models in

Table 1: **Comparison to prior work on *mini*-ImageNet.** Average 5-way classification accuracies (%) on *mini*-ImageNet meta-test split are reported with 95% confidence intervals. Results of prior works are adopted from Lee et al. (2019) and original papers. [†] used extra cross-modal information.

| Model | Backbone | *mini*-ImageNet 5-way | |
| --- | --- | --- | --- |
| | | 1-shot | 5-shot |
| Meta-Learning LSTM (Ravi & Larochelle, 2016) | Conv-4 | $43.44 \pm 0.77$ | $60.60 \pm 0.71$ |
| Matching Networks (Vinyals et al., 2016) | Conv-4 | $43.56 \pm 0.84$ | $55.31 \pm 0.73$ |
| Prototypical Networks (Snell et al., 2017) | Conv-4 | $49.42 \pm 0.78$ | $68.20 \pm 0.66$ |
| Transductive Prop Nets (Liu et al., 2018) | Conv-4 | $55.51 \pm 0.86$ | $69.86 \pm 0.65$ |
| MetaOptNet (Lee et al., 2019) | Conv-4 | $52.87 \pm 0.57$ | $68.76 \pm 0.48$ |
| Negative Margin (Liu et al., 2020) | Conv-4 | $52.84 \pm 0.76$ | $70.41 \pm 0.66$ |
| ConstellationNet (ours) | Conv-4 | $\mathbf{58.82 \pm 0.23}$ | $\mathbf{75.00 \pm 0.18}$ |
| SNAIL (Mishra et al., 2018) | ResNet-12 | $55.71 \pm 0.99$ | $68.88 \pm 0.92$ |
| TADAM (Oreshkin et al., 2018) | ResNet-12 | $58.50 \pm 0.30$ | $76.70 \pm 0.30$ |
| TapNet (Yoon et al., 2019) | ResNet-12 | $61.65 \pm 0.15$ | $76.36 \pm 0.10$ |
| Variational FSL (Zhang et al., 2019) | ResNet-12 | $61.23 \pm 0.26$ | $77.69 \pm 0.17$ |
| MetaOptNet (Lee et al., 2019) | ResNet-12 | $62.64 \pm 0.61$ | $78.63 \pm 0.46$ |
| CAN (Hou et al., 2019) | ResNet-12 | $63.85 \pm 0.48$ | $79.44 \pm 0.34$ |
| SLA-AG (Lee et al., 2020) | ResNet-12 | $62.93 \pm 0.63$ | $79.63 \pm 0.47$ |
| Meta-Baseline (Chen et al., 2020) | ResNet-12 | $63.17 \pm 0.23$ | $79.26 \pm 0.17$ |
| AM3 (Xing et al., 2019) [†] | ResNet-12 | $65.21 \pm 0.30$ | $75.20 \pm 0.27$ |
| ProtoNets + TRAML (Li et al., 2020) | ResNet-12 | $60.31 \pm 0.48$ | $77.94 \pm 0.57$ |
| AM3 + TRAML (Li et al., 2020) [†] | ResNet-12 | $\mathbf{67.10 \pm 0.52}$ | $79.54 \pm 0.60$ |
| Negative Margin (Liu et al., 2020) | ResNet-12 | $63.85 \pm 0.81$ | $\mathbf{81.57 \pm 0.56}$ |
| ConstellationNet (ours) | ResNet-12 | $64.89 \pm 0.23$ | $79.95 \pm 0.17$ |

Table 2: **Comparison to prior work on FC100 and CIFAR-FS.** Average 5-way classification accuracies (%) on CIFAR-FS and FC100 meta-test split are reported with 95% confidence intervals. Results of prior works are adopted from Lee et al. (2019) and original papers.

| Model | Backbone | CIFAR-FS 5-way | | FC100 5-way | |
| --- | --- | --- | --- | --- | --- |
| | | 1-shot | 5-shot | 1-shot | 5-shot |
| MAML (Finn et al., 2017) | Conv-4 | $58.9 \pm 1.9$ | $71.5 \pm 1.0$ | - | - |
| Prototypical Networks (Snell et al., 2017) | Conv-4 | $55.5 \pm 0.7$ | $72.0 \pm 0.6$ | - | - |
| Relation Networks (Sung et al., 2018) | Conv-4 | $55.0 \pm 1.0$ | $69.3 \pm 0.8$ | - | - |
| R2D2 (Bertinetto et al., 2018) | Conv-4 | $65.3 \pm 0.2$ | $79.4 \pm 0.1$ | - | - |
| SIB (Hu et al., 2020) | Conv-4 | $68.7 \pm 0.6$ | $77.1 \pm 0.4$ | - | - |
| ConstellationNet (ours) | Conv-4 | $\mathbf{69.3 \pm 0.3}$ | $\mathbf{82.7 \pm 0.2}$ | - | - |
| Prototypical Networks (Snell et al., 2017) | ResNet-12 | $72.2 \pm 0.7$ | $83.5 \pm 0.5$ | $37.5 \pm 0.6$ | $52.5 \pm 0.6$ |
| TADAM (Oreshkin et al., 2018) | ResNet-12 | - | - | $40.1 \pm 0.4$ | $56.1 \pm 0.4$ |
| MetaOptNet-RR (Lee et al., 2019) | ResNet-12 | $72.6 \pm 0.7$ | $84.3 \pm 0.5$ | $40.5 \pm 0.6$ | $55.3 \pm 0.6$ |
| MetaOptNet-SVM (Lee et al., 2019) | ResNet-12 | $72.0 \pm 0.7$ | $84.2 \pm 0.5$ | $41.1 \pm 0.6$ | $55.5 \pm 0.6$ |
| ConstellationNet (ours) | ResNet-12 | $\mathbf{75.4 \pm 0.2}$ | $\mathbf{86.8 \pm 0.2}$ | $\mathbf{43.8 \pm 0.2}$ | $\mathbf{59.7 \pm 0.2}$ |

Appendix A.8. We observe ConstellationNet with negative margin brings 0.52% improvement to ConstellationNet, and obtains 6.93% gain compared with baseline on *mini*-ImageNet.

# 6 MODEL ANALYSIS

## 6.1 ARCHITECTURE ALTERNATIVES

In Table 3, we first study the role of each module in ConstellationNet, where the number of parameters is controlled approximately equivalent to the baseline's size. Our constellation model brings 6.41% and 2.59% improvements over baseline on 1-shot Conv-4 and ResNet-12 results. Combined with our multi-branch training procedure, the model further improves additional 1.34% and 1.26% on 1-shot Conv-4 and ResNet-12, respectively. Finally, feature augmentation from penultimate layer to final output embedding brings additional 0.45% and 0.27% improvements on two variants.

We also test the baseline model with extra channels in the Table 3. The new model only shows slight improvements over original baseline, and is outperformed by our ConstellationNet with a large margin. We also obtain WRN-28-10 baseline results to validate our improvement. While making ResNet baselines deeper and wider, our ConstellationNet still outperforms this strong baseline. In Figure 2 (e), we further study whether the performance gap between ConstellationNet and baseline can be reduced by simply altering the baseline's model complexity using e.g. more convolution channels. Although the trend of baseline accuracy increases when increasing the model parameter number gradually, the performance gap is still significant. This validates our concept that modeling hierarchical part structures can greatly benefit features learned from convolution operation, and obtain a more robust feature representation. In addition, applying self-attention on the distance map (6-th

Table 3: **Effectiveness of modules.** Average classification accuracies (%) on *mini*-ImageNet meta-test split. We compare our ConstellationNet with alternative architectures including the baseline and the modified baseline with extra channels based on Conv-4 and ResNet-12. We also include a baseline with WideResNet-28-10 (Zagoruyko & Komodakis, 2016) backbone for comparison.

| Baseline | Cell Feature Clustering | Cell Relation Modeling | Multi Branch | Feature Augment | Extra Channels | 1x1 Convolution | #Params Conv-4/Res-12 | Conv-4 1-shot | Conv-4 5-shot | ResNet-12 1-shot | ResNet-12 5-shot |
|---|---|---|---|---|---|---|---|---|---|---|---|
| ✓ | | | | | | | 117K/8.0M | 50.62 ± 0.23 | 68.40 ± 0.19 | 60.77 ± 0.22 | 78.76 ± 0.17 |
| ✓ | | | | | ✓ | | 222K/16M | 51.76 ± 0.22 | 69.54 ± 0.18 | 61.45 ± 0.22 | 79.33 ± 0.16 |
| ✓ | ✓ | | | | | | 146K/8.3M | 53.34 ± 0.23 | 70.61 ± 0.19 | 62.24 ± 0.23 | 79.55 ± 0.16 |
| ✓ | | ✓ | | | | | 184K/9.7M | 55.92 ± 0.23 | 73.02 ± 0.18 | 62.75 ± 0.23 | 79.21 ± 0.17 |
| ✓ | | ✓ | | | | ✓ | 192K/8.4M | 55.46 ± 0.23 | 72.52 ± 0.18 | 61.54 ± 0.24 | 76.51 ± 0.18 |
| ✓ | ✓ | ✓ | | | | | 200K/8.4M | 57.03 ± 0.23 | 74.09 ± 0.18 | 63.36 ± 0.23 | 79.72 ± 0.17 |
| ✓ | ✓ | ✓ | ✓ | | | | 200K/8.4M | 58.37 ± 0.23 | 74.52 ± 0.18 | 64.62 ± 0.23 | 79.60 ± 0.17 |
| ✓ | ✓ | ✓ | ✓ | ✓ | | | 200K/8.4M | **58.82 ± 0.23** | **75.00 ± 0.18** | **64.89 ± 0.23** | **79.95 ± 0.17** |
| | | | | | | | **WRN** | | | **WideResNet-28-10** | |
| ✓ | | | | | ✓ | | 36.5M | | | 61.54 ± 0.25 | 79.41 ± 0.23 |

row: 57.03% on Conv-4, 1-shot) achieves better performance than directly applying it to the original cell features (i.e. convolutional feature map) (4-th row: 55.92% on Conv-4, 1-shot). We also tried to replace the cell feature clustering module with a 1x1 convolution layer (output dimension is equal to the number of clusters) (5-th row: 55.46% on Conv-4, 1-shot). It is worse than our results (6-th row) as well. We observe that the 1x1 convolution layer is less expressive than the cell feature clustering module, making it difficult to extract enough context information during cell relation modeling.

## 6.2 Modules Analysis

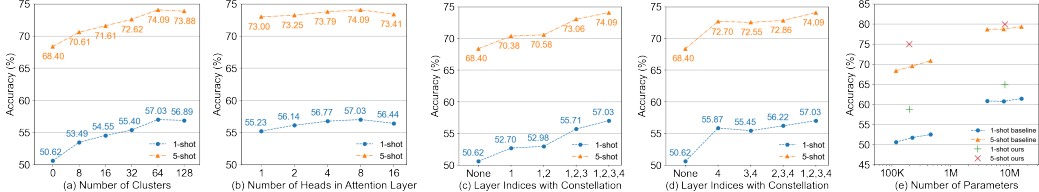

Figure 2: **Modules analysis.** (a, b, c, d) We study the effectiveness of changing the number of clusters, the number of heads in attention layer, and the layer indices with constellation based on Conv-4, (e) We demonstrate the performance gain of our ConstellationNet is unmatched by increasing the model complexity of our baselines. All experiments are done on *mini*-ImageNet.

In Figure 2 (a), we vary the number of clusters adapted in all layers to observe the performance change. We found that increasing the number of clusters improves the accuracy in general, and set clusters to 64 is optimal in terms of both model size and classification performance. Figure 2 (b) shows the number of attention heads does not effect performance as much as the number of cluster, and 8-head attention obtains 1.80% performance gain on the 1-shot setting compared to 1-head attention. In Figure 2 (c, d), we also study the effectiveness of clustering algorithm applied to different layers. The results show both early features and high-level features benefit from introducing clusters algorithm into the original CNN architecture.

## 6.3 Visualization

Figure 3 demonstrates the visualization of cluster centers in each layer of Conv-4 model on *mini*-ImageNet. In the upper part of the figure, each image shows patches corresponding to the nearest cell features to a cluster center (i.e. with lowest Euclidean distance). It is observed that clusters in early layers (e.g. layer 1,2) represent simple low-level patterns while the clusters in high layers (e.g. layer 3,4) indicate more complex structures and parts. In the lower part of the figure, we choose two cluster centers from layer 4 for further interpretation: The left one with green box could possibly represent *legs* since it consists of various types of legs from human, dog and other animals. The right one with the red box shows most nearest cell features to this cluster center are parts with bird's head or beetles, which share a dotted structure (i.e. black dots on beetles / eyes on bird's head).

The left side of Figure 4 shows the visualization of cell features that are assigned to different clusters. For each image, we extract the assignment maps corresponding to three cluster centers generated in the last constellation module of Conv-4 and find multiple cell features with the highest assignments within each assignment map. The locations of cell features are projected back in the original image space, marked by three different colors of "·" in the raw image to show three different feature clusters. For a given class of images, the same cluster centers are selected for comparison across 6 samples. As shown in Figure 4, we observe part information of each class is explicitly discovered. For the bird

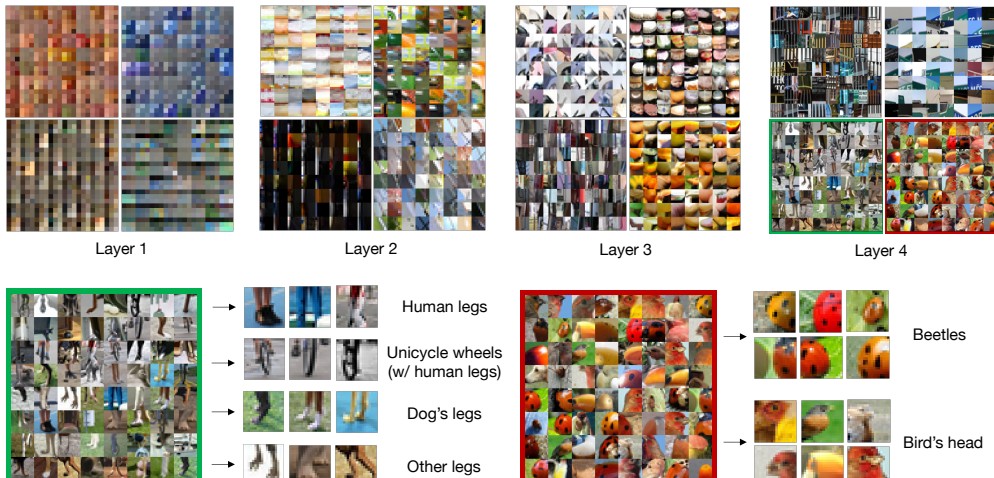

Figure 3: **Visualization of cluster centers.** (Upper) We visualize four cluster centers in each layer by showing patches associated with cell features that have the nearest distance to the clustering center. (Lower) Identifying parts from two cluster centers in layer 4: Left one with green box represents various types of legs. Right one with red box mostly shows beetles and bird's head, sharing a dotted structure.

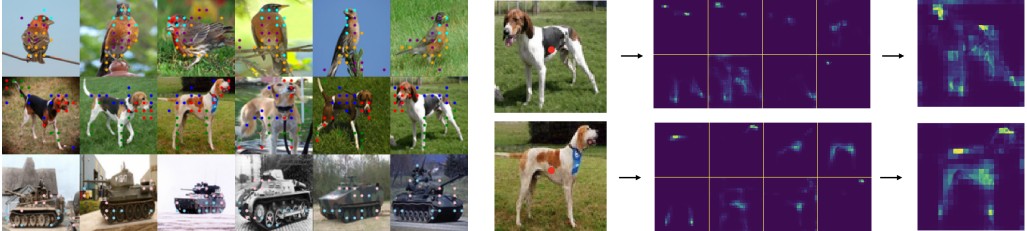

Figure 4: **Visualization of the cells assignment and attention maps.** (Left) Each color represents a cluster, and each point, marked as "·", represents a cell assigned to a cluster center. We demonstrate 6 samples for each class (bird, dog and tank). (Right) We visualize attention maps of one query feature (at the location of red point in left part) with all key features. The middle part shows the attention maps corresponding to 8 heads in the multi-head attention. The right part shows an overlapped map of all attention maps.

category, we can see different parts in each image, including head (cyan "·"), body (purple "·") and tail (yellow "·"). For the dog category, we see parts including heads (red "·"), legs (green "·") and body (blue "·"). For the tank category, we see parts like track (light blue "·") and turret (pink "·").

The right side of Figure 4 visualizes the attention maps in the cell relation model. We use the last constellation module in the ResNet-12 model for visualization since it captures high-level features that better represent parts. We choose one query feature at the center of the object and show its attention map to all key features. The middle part of the figure shows the attention maps corresponding to 8 heads in the multi-head attention. It is observed that some parts are identified such as head (second map in first row), legs (first two map in second row), buttock (first map in first row) and body (second map in the second row). A merged attention map by overlaying all 8 attention maps is presented at right part of the figure. It indicates that all the attention heads together can extract the features of the whole object, which would be useful for final classification.

# 7 CONCLUSION

In this paper, we present ConstellationNet by introducing an explicit feature clustering procedure with relation learning via self-attention. We implement a mini-batch soft $k$-means algorithm to capture the cell feature distribution. With integrated implicit (standard CNN modules) and explicit (cell feature clustering + cell relation modeling) representations, our proposed ConstellationNet achieves significant improvement over the competing methods on few-shot classification benchmarks.

ACKNOWLEDGMENTS

This work is funded by NSF IIS-1618477 and NSF IIS-1717431. We thank Qualcomm Inc. for an award support. We thank Kwonjoon Lee, Tiange Luo and Hao Su for valuable feedbacks.

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

# A   APPENDIX

## A.1   FEW-SHOT LEARNING FRAMEWORK

In this section, we introduce background concepts of meta-learning and elaborate the few-shot learning framework used in our ConstellationNet.

**Meta-Learning in Few-Shot Classification.**   Current few-shot learning is typically formulated as a *meta-learning* task (Finn et al., 2017), in which an dataset $\mathcal{D}^{\text{base}}$ is used to provide commonsense knowledge and a dataset $\mathcal{D}^{\text{novel}}$ for the few-shot classification. $\mathcal{D}^{\text{base}}$ has the classes $\mathcal{C}_{\text{base}}$ which are disjoint from the $\mathcal{C}_{\text{novel}}$ in $\mathcal{D}^{\text{novel}}$ to ensure fairness. There are two stages, *meta-training* and *meta-test*, in the meta-learning framework: In *meta-training* stage, we attempt to train a model to learn generic features from $\mathcal{D}^{\text{base}}$. In *meta-test* stage, we adapt the model on the limited training split from $\mathcal{D}^{\text{novel}}$ and evaluate the performance of the model on the test split.

**ProtoNet-Based Framework.**   In our ConstellationNet, we adopt ProtoNet (Snell et al., 2017) as the base few-shot learning framework. In ProtoNet, the dataset $\mathcal{D}^{\text{novel}}$ is represented by a series of $K$-way $N$-shot tasks $\{\mathcal{T}\}$ where each task consists of a *support* set and a *query* set, i.e. $\mathcal{T} = (\mathcal{T}^{\text{supp}}, \mathcal{T}^{\text{query}})$. The support set $\mathcal{T}^{\text{supp}}$ contains $K$ classes and each class has $N$ examples from the training split of $\mathcal{D}^{\text{novel}}$, which are used to adapt the model in *meta-test* stage. The query set $\mathcal{T}^{\text{query}}$ from the test split of $\mathcal{D}^{\text{novel}}$ is then used to evaluate the model.

The ProtoNet attempts to learn a generic feature extractor $\phi(\mathbf{x})$ on image $\mathbf{x}$, and represent a class $k$ by the prototype $\mathbf{c}_k$, which is the average feature of examples from support set $\mathcal{T}^{\text{supp}}$ with this class:

$$\mathbf{c}_k = \frac{1}{|N|} \sum_{(\mathbf{x},y) \in \mathcal{T}^{\text{supp}}, y=k} \phi(\mathbf{x}) \tag{11}$$

During the *meta-test* stage, we use the prototypes to compute the probability $p_k$ of a query example $\mathbf{x}' \in \mathcal{T}^{\text{query}}$ on class $k$ and predict its label $y'$:

$$p_k = p(y = k | \mathbf{x}', \mathcal{T}^{\text{supp}}) = \frac{\exp(d(\mathbf{x}', \mathbf{c}_k))}{\sum_{k'} \exp(d(\mathbf{x}', \mathbf{c}_{k'}))}, \quad y' = \arg\max_k p_k. \tag{12}$$

where $d(\cdot, \cdot)$ is a cosine similarity function (different from the Euclidean distance in Snell et al. (2017)).

During the *meta-training* stage, there are two different training schemes: The *prototypical scheme* from ProtoNet uses an *episodic learning* strategy that also formulates the dataset $\mathcal{D}^{\text{base}}$ as a series of tasks $\{\mathcal{T}\}$. The negative log-likelihood loss $\mathcal{L}(\phi)$ is optimized:

$$\ell(\mathcal{T}^{\text{supp}}, \mathcal{T}^{\text{query}}) = \mathbb{E}_{(\mathbf{x}', y') \in \mathcal{T}^{\text{query}}} - \log p(y = y' | \mathbf{x}', \mathcal{T}^{\text{supp}}), \tag{13}$$

$$\mathcal{L}(\phi) = \mathbb{E}_{\mathcal{T} = (\mathcal{T}^{\text{supp}}, \mathcal{T}^{\text{query}}) \sim \mathcal{D}^{\text{base}}} \ell(\mathcal{T}^{\text{supp}}, \mathcal{T}^{\text{query}}). \tag{14}$$

Another way is the *standard classification scheme* (Chen et al., 2020): It simply uses $\mathcal{D}^{\text{base}}$ as a standard classification dataset $\{(\mathbf{x}, y)\}$ consisting of $Q$ classes in total. Thus, a cross-entropy loss $\mathcal{L}(\phi)$ is optimized:

$$\mathcal{L}(\phi) = \mathbb{E}_{(\mathbf{x},y) \sim \mathcal{D}^{\text{base}}} - \log \frac{\exp(\mathbf{w}_y \cdot \phi(\mathbf{x}))}{\sum_q \exp(\mathbf{w}_q \cdot \phi(\mathbf{x}))} \tag{15}$$

where $\mathbf{w}_q$ is the linear weight for class $q$. In our ConstellationNet, we use the standard classification scheme at default. For the experiment with multi-branch network, we use the prototypical scheme and standard classification scheme for separate branches.

## A.2   DATASETS

The CIFAR-FS dataset (Bertinetto et al., 2018) is a few-shot classification benchmark containing 100 classes from CIFAR-100 (Krizhevsky et al., 2009). The classes are randomly split into 64, 16 and 20 classes as meta-training, meta-validation and meta-testing set respectively. For each class, it

contains 600 images of size $32 \times 32$. We adopt the split from Lee et al. (2019). The FC100 dataset (Oreshkin et al., 2018) is another benchmark based on CIFAR-100 where classes are grouped into 20 superclasses to void the overlap between the splits. The *mini*-ImageNet dataset (Vinyals et al., 2016) is a common benchmark for few-shot classification containing 100 classes from ILSVRC-2012 (Deng et al., 2009). The classes are randomly split into 64, 16 and 20 classes as meta-training, meta-validation and meta-testing set respectively. For each class, it contains 600 images of size $84 \times 84$. We follow the commonly-used split in Ravi & Larochelle (2016), Lee et al. (2019) and Chen et al. (2020). In all experiments, we conduct data augmentation for the meta-training set of all datasets to match Lee et al. (2019)'s implementation.

### A.3 NETWORK BACKBONE

*Conv-4.* Following Lee et al. (2019), we adopt the same network with 4 convolutional blocks. Each of the 4 blocks has a $3 \times 3$ convolutional layer, a batch normalization layer, a ReLU activation and a $2 \times 2$ max-pooling layer sequentially. The numbers of filters are 64 for all 4 convolutional layers.

*ResNet-12.* Following Chen et al. (2020), we construct the residual block with 3 consecutive convolutional blocks followed by an addition average pooling layer where each convolutional block has a $3 \times 3$ convolutional layer, a batch normalization layer, a leaky ReLU activation, and max-pooling layers. The ResNet-12 network has 4 residual blocks with each filter size set to 64, 128, 256, 512, respectively.

*WRN-28-10.* WideResNet expands the residual blocks by increasing the convolutional channels and layers (Zagoruyko & Komodakis, 2016). WRN-28-10 uses 28 convolutional layers with a widening factor of 10.

### A.4 CONSTELLATION MODULE CONFIGURATION

To achieve the best performance with constellation modules, we do not always fully enable them after all the convolutional layers. For Conv-4, we use constellation modules after all four convolutional layers, but the cell relation modeling module is disabled in first two constellation modules due to the high memory consumption. For ResNet-12, we enable the constellation modules after the convolutional layer 1,7,8,9 and disable the relation modeling module in the first constellation module. We use the deep supervision in ResNet-12 to stablize the training of constellation modules.

### A.5 SELF-ATTENTION SETTINGS

We follow the common practice in Vaswani et al. (2017) to set the attention layer with residual connections, dropout and layer normalization. The sine positional encoding follows settings in Carion et al. (2020).

### A.6 TRAINING DETAILS

*Optimization Settings.* We follow implementation in Lee et al. (2019), and use SGD optimizer with initial learning rate of 1, and set momentum to 0.9 and weight decay rate to $5 \times 10^{-4}$. The learning rate reduces to 0.06, 0.012, and 0.0024 at epoch 20, 40 and 50. The inverse temperature $\beta$ is set to 100.0 in the cluster assignment step, and $\lambda$ is set to 1.0 in the centroid movement step.

### A.7 ABLATION STUDY ON THE NUMBER OF CLUSTERS

Table 4 studies the number of clusters needed for random and similar classes. The result shows the optimal number of clusters are less affected by the number of clusters but more affected by the similarity between classes. Less number of clusters are needed for dataset with classes of high similarity, which aligns with our intuition, limited number of patterns exist in this dataset so that small number of clusters are enough to represent its part-based information.

FC100 training dataset consists of 60 classes that are grouped evenly into 12 superclasses. In the random classes group, the training dataset includes 6 randomly selected super-classes (i.e., 30 classes) and models are trained with 8, 16, 32, 64 and 128 number of clusters. The highest accuracy occurs at 16 clusters (1-shot: 39.12% in ResNet-12). In the similar classes group, 30 classes are randomly

Table 4: **Ablation study on the number of clusters for random and similar classes.** We investigate how similarities of images in the training dataset affect the optimal number of clusters. The first group of experiments use training dataset with 30 similar classes while the second group use 30 random classes from FC100 dataset, all of which performed on ResNet-12 with Constellation module.

| # Clusters | Similar Classes | | Random Classes | |
|---|---|---|---|---|
| | 1-shot | 5-shot | 1-shot | 5-shot |
| 8 | $38.9 \pm 0.2$ | $\mathbf{52.8 \pm 0.2}$ | $40.9 \pm 0.2$ | $54.5 \pm 0.2$ |
| 16 | $\mathbf{39.1 \pm 0.2}$ | $51.8 \pm 0.2$ | $40.9 \pm 0.2$ | $\mathbf{54.9 \pm 0.2}$ |
| 32 | $38.7 \pm 0.2$ | $52.3 \pm 0.2$ | $40.9 \pm 0.2$ | $54.7 \pm 0.2$ |
| 64 | $38.8 \pm 0.2$ | $52.3 \pm 0.2$ | $\mathbf{41.2 \pm 0.2}$ | $\mathbf{54.9 \pm 0.2}$ |
| 128 | $38.8 \pm 0.2$ | $52.1 \pm 0.2$ | $40.8 \pm 0.2$ | $54.7 \pm 0.2$ |

sampled from the original training dataset and we repeat the same experiments as above. The highest accuracy occurs at 64 clusters (1-shot: 41.22% in ResNet-12), which is much more than the 16 clusters used for images from similar classes.

## A.8 ADDITIONAL EXPERIMENTS WITH NEGATIVE MARGIN

Table 5: **Additional experiments with the use of negative margin.** Average classification accuracies (%) on *mini*-ImageNet meta-test split. We compare our ConstellationNet and baseline with and without the negative margin loss based on Conv-4.

| Baseline | Cell Feature Clustering | Cell Relation Modeling | Negative Margin | Conv-4 | |
|---|---|---|---|---|---|
| | | | | 1-shot | 5-shot |
| ✓ | | | | $50.62 \pm 0.23$ | $68.40 \pm 0.19$ |
| ✓ | | | ✓ | $51.42 \pm 0.23$ | $68.84 \pm 0.19$ |
| ✓ | ✓ | ✓ | | $57.03 \pm 0.23$ | $74.09 \pm 0.18$ |
| ✓ | ✓ | ✓ | ✓ | $57.55 \pm 0.23$ | $74.49 \pm 0.18$ |

Table 5 studies the use of negative margin loss (Liu et al., 2020) on our Conv-4 models. In the negative margin loss, we use the inner-product similarity, the temperature coefficient $\beta = 1.0$ and the negative margin $m = -0.5$, which attains the best performance improvement on our models. Besides, we do not have the fine-tune step during meta-test. Our baseline with the negative margin loss obtains 0.80% improvement on 1-shot and 0.44% improvement on 5-shot compared with the baseline. Similarly, our ConstellationNet with the negative margin loss achieves 0.52% improvement on 1-shot and 0.40% improvement on 5-shot. The consistent improvement of negative margin loss on the baseline and our ConstellationNet indicates that our constellation module is orthogonal to the negative margin loss, and both modules can boost the performance on few-shot classification.

## A.9 CLARIFICATION ON CLUSTERING PROCEDURE

In this section, we add more clarification on our cell feature clustering procedure in Sec. 4.1: During the training stage, the global cluster centers $\mathcal{V} = \{\mathbf{v}_k\}$ are updated by the computed clustering centers $\{\mathbf{v}'_k\}$ in current mini-batch. Each update to a cluster center $\mathbf{v}_k$ is weighted by a momentum coefficient $\eta$ determined by the value of an associated counter $s_k$, since we would like to avoid large adjustment from the current mini-batch in order to stabilize the global cluster centers. Besides, the mini-batches of examples are randomly drawn from the dataset following Sculley (2010), without specialized design to optimize clustering learning. During the evaluation stage, we fix the global cluster centers $\mathcal{V}$ in the forward step of our model, avoiding the potential information leak or transduction from the test mini-batches.

## A.10 MULTI-BRANCH DETAILS

Our embedding $\phi(\mathbf{x})$ is separated into two branches after a shared stem (Y-shape), which is defined as $\phi(\mathbf{x}) = \{\phi^{cls}(\mathbf{x}), \phi^{proto}(\mathbf{x})\}$ and $\phi^{cls}(\mathbf{x}) = g^{cls}(f^{stem}(\mathbf{x}))$, $\phi^{proto}(\mathbf{x}) = g^{proto}(f^{stem}(\mathbf{x}))$. Two branches $\phi^{cls}(\mathbf{x}), \phi^{proto}(\mathbf{x})$ are trained by standard classification and prototypical schemes separately

in a multi-task learning fashion. During the testing time, $\phi^{\mathrm{cls}}(\mathbf{x})$ and $\phi^{\mathrm{proto}}(\mathbf{x})$ are concatenated together to compute distance between support prototypes and query images.

For our ConstellationNet, we split the network into two branches after the second convolutional blocks (Conv-4) or the second residual blocks (ResNet-12). We keep the shared stem identical to the network backbone and reduce the channels of two separate branches to match the parameter size of the model without multi-branch.

### A.11 CONNECTION WITH CAPSULE NETWORKS

A notable development to learning the explicit structured representation in an end-to-end framework is the capsule networks (CapsNets) (Sabour et al., 2017). The line of works on CapsNets (Sabour et al., 2017; Hinton et al., 2018; Kosiorek et al., 2019; Tsai et al., 2020) intends to parse a visual scene in an interpretable and hierarchical way. Sabour et al. (2017) represents parts and objects in vector-based capsules with a dynamic routing mechanism. Tsai et al. (2020) uses a stacked autoencoder architecture to model the hierarchical relation among parts, objects and scenes. Here our ConstellationNet maintains part modeling by enabling the joint learning of the convolution and constellation modules to simultaneously attain implicit and explicit representations.

