# OpenReview forum: "Attentional Constellation Nets for Few-Shot Learning"
_ICLR.cc/2021/Conference — ICLR 2021 Poster_

### Official Review · AnonReviewer2 · 2020-10-28

**Rating:** 5
**Confidence:** 5

**Review:**

The proposed contellation module is an improved version of non-local block [a], which contains a cell feature clustering module and a self-attention module for modeling pixel-wise (cell-wise) relationships. Inserting this block to the backbones could improve the performance for few-shot learning setting.

Concerns:
1. In my mind, the most important difference between the proposed constellation module and non-local block is the cell feature clustering module. (Multi-head design is widely-used in the extensions of transformer model) Hence, it is important to prove that the newly inserted cell feature clustering module is crucial in the constellation module at least in the few-shot learning setting. It would be bonus to do comparison in other important benchmarks.

2. The corresponding intuition that why the cell feature clustering module works well is also needed. For example, why should we need a shared part patterns (cluster centers) for all different images. Is 128 shared centers enough for all data points? It would be great to visualize the 128 learnt part patterns, e.g. visualizing nearest cell features in the all images.

3. For few-shot learning, there are several papers published recently with state-of-the-art results [b,c], which should be compared in the literature.

[a] Non-local Neural Networks, CVPR 2018
[b] Negative Margin Matters: Understanding Margin in Few-shot Classification, ECCV 2020
[c] Boosting Few-Shot Learning With Adaptive Margin Loss, CVPR 2020

---

> ### Author Response · Authors · 2020-11-20
> **We thank reviewer 2 for the valuable comments. Please see our answers to the questions below.**
>
> We thank reviewer 2 for the valuable feedback. Below please find our answers.
>
> Q1. "In my mind, the most important difference between the proposed constellation module and non-local block is the cell feature clustering module. (Multi-head design is widely-used in the extensions of transformer model) Hence, it is important to prove that the newly inserted cell feature clustering module is crucial in the constellation module at least in the few-shot learning setting. It would be bonus to do comparison in other important benchmarks."
>
> A1. Thank you for pointing out the non-local block paper (Wang et al. 2018) in which the relationship between non-local block and self-attention has been described.
>
> In the paper, we emphasize two highlights of our ConstellationNet model: 1) parts (tokenized representation by the clustering map) and 2) spatial relationship (through positional encoding) among parts. Our novelty comes from fact the parts + their spatial relationship are conveniently modeled via cell feature clustering and the self-attention mechanism in cell relation modeling.
>
> There is a large difference between our work and that of Wang et al. 2018. In non-local neural network (Wang et al. 2018), not only is clustering/codebook learning absent, there is no positional encoding that encodes spatial configuration involved in the method. In this sense, the non-local neural network work (Wang et al., 2018) differs from our in multiple aspects including:
> 1) Motivation (theirs is to capture long-distance dependency beyond local operations vs. ours to model the parts and their spatial relations);
> 2) Implementation (theirs is to have self-attention on raw flattened CNN features vs. ours to have self-attention on clustering map + positional encoding);
> 3) Modules (no clustering and no positional/spatial encoding in Wang et al. 2018);
> 4) Domain (video classification vs. static images for few-shot learning).
>
> The main contribution of our work is a constellation module that performs cell clustering (codebook learning), followed by self-attention on the distance vector (learned code) + positional encoding for each cell.
>
> Please also see Section 4.4 in the updated version for more explanations and explained intuitions.
>
> Additional experiments:
>
> In addition, we conduct an additional ablation study on the effectiveness of the cell feature clustering module in the Table 3 (revised). Details see A1 response to Reviewer1. In short, we have following findings:
> Applying self-attention on the distance map (1-shot: 57.03% in Conv-4) achieves better performance than directly applying it to the raw cell features (i.e. convolutional feature map) (1-shot: 55.92 % in Conv-4) , and better than replacing the cell feature clustering module with a 1x1 convolution layer (output dimension is equal to the number of clusters)  (1-shot: 55.46% in Conv-4).
>
>
> Q2. "The corresponding intuition that why the cell feature clustering module works well is also needed. For example, why should we need a shared part patterns (cluster centers) for all different images. Is 128 shared centers enough for all data points? It would be great to visualize the 128 learnt part patterns, e.g. visualizing nearest cell features in the all images."
>
>
>  A2. Thank you for pointing this out.
>
> Despite the effectiveness of the CNN models in standard supervised image classification, their transfer abilities are limited when training data for novel classes is scarce in few-shot learning tasks. We take the inspiration from the traditional constellation models, and come up with a realization that improves CNN’s modeling capability.
>
> Analogous to classical constellation models (Fergus et al., 2003; Felzenszwalb & Huttenlocher, 2005) to extract parts with their joint spatial relationships, we perform codebook learning by cell clustering followed by self-attention on the code/clustering map and positional encoding. Seamlessly integrating constellation models with convolution operations improves the generalization in representation over the baseline CNNs for few-shot learning.
>
> Please see Section 4.4 in the updated version for more explanations and explained intuitions why the cell feature clustering module works well.
>
> Taking your advice, we visualize nearest cell features in images.  Please see Figure 3 (revised paper) to see part patterns. Particularly, we observe features belonging to the same part concept (legs) are grouped to the same cluster even though they belong to different classes (human and dog). This explains why only limited clusters are needed to gain our performance boost.
>
> Please see Figure 4 (revised paper) to see visualization of the cells assignment and attention maps. The left figure shows how cells are assigned to different clusters, and right figure shows how the spatial relationship between parts (the learned code/distance vector from clustering) are captured.

---

> > ### Author Response · Authors · 2020-11-20
> > **Response to reviewer 2 (Cont.)**
> >
> > Q3. "For example, why should we need a shared part patterns (cluster centers) for all different images."
> >
> > A3. Sharing part patterns (cluster centers) is one of the reasons for the effectiveness in part-based (bag of words) models where the parts (BoW) become an abstraction, and hence achieving effective generalization and representation capability, especially when training data is limited.
> >
> > Q4. "For few-shot learning, there are several papers published recently with state-of-the-art results [b,c], which should be compared in the literature."
> >
> >  A4. Thank you for bringing this up. We have included results from the two papers in Table 1 (Revised). Our ConstellationNet is orthogonal to the mentioned marginal loss-based methods, and we also do not use extra cross-modal information: AM3 (Xing et al, 2019) manages to adaptively combine language and visual information to improve classification accuracy. On top of AM3, AM3 TRAML (Li et al, 2020) emphasizes on leveraging semantic similarity among different classes to generate adaptive margin among classes. Negative Margin (Liu et al, 2020) improves the performance by introducing a negative margin loss to make the learned metrics more discriminative to novel classes.  These methods either depend on the margin loss to guide the training, or introduce fusion procedure with language information to provide stronger prior knowledge and richer context information. On the contrary, our model enhances the embedding generalization ability by incorporating its own part-based representation.
> >
> >
> > For the lightweight backbone (Conv-4), our model outperforms Negative Margin (Liu et al, 2020) by a large margin (5.98% for 1-shot, 4.59% for 5-shot). For the backbone like ResNet-12, our model still achieves competitive results: Our performance is better than AM3 TRAML (Li et al, 2020) (with extra cross-modal information) in the 5-shot setting but lower in the 1-shot setting; Our performance is better than Negative Margin (Liu et al, 2020) in 1-shot setting but not in 5-shot setting.

---

> > > ### Author Response · Authors · 2020-11-25
> > > **Response to reviewer 2 (Cont.)**
> > >
> > > (cont.) A4. Additional Experiments With Negative Margin
> > >
> > > To verify the orthogonality between our method and existing approaches, we adapt the negative margin loss following Negative Margin (Liu et al, 2020) to our Conv-4 models in Appendix A.7. On mini-ImageNet benchmark, we observe that our baseline with the negative margin loss (51.42%) obtains 0.80% improvement on 1-shot compared with the baseline (50.62%). Similarly, our ConstellationNet with the negative margin loss (57.55%) achieves 0.52% improvement on 1-shot compared with our ConstellationNet (57.03%). The consistent improvement of negative margin loss on the baseline and our ConstellationNet indicates that our constellation module is orthogonal to the negative margin loss, and both modules can boost the performance on few-shot classification.

---

### Official Review · AnonReviewer3 · 2020-10-28
**Promising idea, paper can be strengthened with more clarifications and justifications**

**Rating:** 6
**Confidence:** 4

**Review:**

*** Update ***

I thank the authors for their very detailed response. Most of my concerns have been addressed and I now recommend acceptance.

***


The authors propose a method for few-shot learning that introduces a new module in the feature encoder of the prototypical network model. Their approach is inspired by constellation models, which represent objects as a set of parts and model their spatial relationships.
The proposed model alternates between standard convolutional layers and a constellation inspired block. This block first clusters image features across a batch, computes a distance map between each feature and cluster center, then obtain a final feature vector via self attention on the distance map.

STRENGTHS

Exploiting object parts and their relationships is a promising direction for few-shot learning, where one may aim to share knowledge about objects structures and common parts. The proposed approach achieves good overall performance, and a large performance gain in small capacity networks when introducing the constellation module.
The experimental analysis of the model and parameters is quite thorough, in particular the experiment evaluating the influence of the extended model capacity.

WEAKNESSES

** missing references **
-First, the literature review should be extended to comment on recent approaches relying on part discovery/recognition. All provided references are 10+  years old, and recent works on compositional and part based representations/models for classification should be discussed.
Few examples could be:
CoupleNet, Zhu et al, ICCV 2017
Object-Part Attention Model for Fine-GrainedImage Classification, IEEE TIP, 2018
Weakly Supervised Complementary Parts Models for Fine-Grained Image Classification from the Bottom Up, CVPR 2019
Exploiting spatial relation for fine-grained image classification, Pattern Recognition, 2019
Learning Compositional Representations for Few-Shot Recognition, ICCV 2019
While these works are not necessarily closely related to the proposed work and do not reevaluate novelty, they are contemporary part based works that explore part based modelling.

More importantly, authors should mention and discuss
‘Neural activation models’ Simon et al 2015, ICCV
Which aims to integrate a constellation model within a neural network model and is therefore strongly related work.

** clarity/justifications **
-While the approach appears to yield good performance increase, it is difficult to comprehend intuitively why that is the case. Justification for certain model decisions is not clearly provided, and certain claims are not supported by evidence. For example, the claim that the clustering procedure explicitly identifies object parts is not obvious not clearly shown in experiments.
Indeed, clusters and their centers are assumed to represent object parts. However, experiments demonstrate that larger numbers of clusters yield stronger performance, suggesting that a coarse superpixel type clustering might be more promising than hoping to identify object parts across images. Similarly, the results in Fig 3 suggest that the discovered clusters are more appearance oriented than identifying parts (cf brown vs white on dog instead of recognising parts e.g. ears).

The clustering process itself lacks clarity. Are cluster centers fixed after training?  Is the optimal number of cluster affected by the number of classes/similarities between classes? Are all cluster centers receiving assignments for each batches? Are batches constructed so as to optimise cluster learning? Can authors expand on the role of count parameter s? Are different clusters relevant to e.g. dog classes different from clusters for bird classes?
As intuitively these clusters represent object parts, providing more attention to these, both in terms of explanation and experiments would be preferable.

-The motivation behind the use of a distance map is unclear. Could authors elaborate on why performing self attention on the distance map provides relevant information vs e.g. self attention on the clustered feature maps?

-The paper would benefit from a clearer depiction of the constellation model and how the proposed approach relates to them, intuitively. With the current writing, the concept of constellation models is quickly brushed over, and the motivation behind the use of a distance map + attention is not clearly stated but only proposed as an alternative to the probabilistic modelling used in old school constellation models. The paper would strongly benefit from providing justification on why this is a valid alternative, and what we hope to learn using this strategy; and more precisely, why this strategy can be viewed as a constellation type model.

-Several aspects of the few-shot learning formulation should be more clearly explained. While the paper is perfectly understandable for a reader accustomed to FSL methods and settings, the lack of explanations regarding episode training, meta-training/testing, and the protonet model itself would make it very difficult to follow for a non expert.

RECOMMENDATION

The proposed work aims to introduce part representation in few-shot models, which is an appealing strategy. Adapting popular traditional models to contemporary settings is a promising idea, and the proposed method reaches SOTA performance on standard benchmark.

The paper in its current state needs a little more attention, and I will be happy to increase my rating if my main concerns are addressed
1-	Relating the model and its components more closely to constellation models, and justification as to why the proposed strategy is a better implementation of constellations in deep learning framework than Simon et al. 2015
2-	Providing clarifications regarding design decisions, experimental setting, and more intuition. In particular regarding the distance map and cluster centers. If possible, according more attention to interpretability/observed behaviour of cluster centers in the experimental section.

ADDITIONAL COMMENTS
-	Experiments are missing important details. For example, it is not specified for experiments in Figure 2 and 3 which dataset and parameter configurations are used. In particular for figure 3, is the number of clusters set to 64? Are all cluster centers relevant to a given class?
-	Examples from the same class provided in this figure look very similar in appearance. What happens when examples of the same class look different? Are same cluster patterns observed?
-	The multi-branch training strategy is not new and was suggested in TADAM, Oreshkin et al., NeurIPS, 2018.
-	Regarding ablation experiments, it would be interesting to see the influence of having a single module on the last layer (where levels of abstraction would be higher) vs modules at every layer.
-	claims regarding ‘explicit modelling of parts’ should be revised. There is no explicit part discovery (nor a guarantee that object parts are indeed discovered), nor a clear, explicit modelling of their interactions. Maybe a more accurate characterisation would be that the approach integrate spatial information between image regions of similar appearance/texture. Similarly, it is not obvious that CNNs extract object parts.

---

> ### Author Response · Authors · 2020-11-20
> **We thank reviewer 3 for the valuable comments. Please see our answers to the questions below.**
>
> We thank reviewer 3 for very detailed and constructive comments. Please see our answers below.
>
> Q1. "Missing references
> CoupleNet, Zhu et al, ICCV 2017
> Object-Part Attention Model for Fine-GrainedImage Classification, IEEE TIP, 2018
> Weakly Supervised Complementary Parts Models for Fine-Grained Image Classification from the Bottom Up, CVPR 2019
> Exploiting spatial relation for fine-grained image classification, Pattern Recognition, 2019
> Learning Compositional Representations for Few-Shot Recognition, ICCV 2019
> Neural Activation Constellations, Simon et al 2015, ICCV"
>
> A1. Thank you for bringing these references to our attention. In the updated version of related work, we have included them in our references, and added discussion with "Neural Activation Constellations: Unsupervised Part Model Discovery with Convolutional Networks", Simon et al 2015, ICCV. For discussion, please see our revised related work and A15 below.
>
> Q2. "Difficult to understand why the model yield good performance intuitively."
>
> A2. Thank you for pointing this out.
>
> Despite the effectiveness of the CNN models in standard supervised image classification, their transfer abilities are limited when training data for novel classes is scarce in few-shot learning tasks. We take the inspiration from the traditional constellation models, and come up with a realization to gain greater modeling capability.
>
> Analogous to classical constellation models (Fergus et al., 2003; Felzenszwalb & Huttenlocher, 2005) to extract parts with their joint spatial relationships, we perform codebook (dictionary) learning by cell clustering followed by self-attention on the code/clustering map and positional encoding for each cell. Seamlessly integrating constellation models with convolution operations improves the generalization in representation over the baseline CNNs for few-shot learning.
>
> Please see Section 4.4 and Section 6.3 in the updated version for more explanations and intuitions.
>
> Q3. "The claim that the clustering procedure explicitly identifies object parts is not obviously not clearly shown in experiments."
>
> A3. We add new visualization in Figure 3 (revised paper). The visualization shows object parts can be identified. As shown in the lower part of the figure, we choose two cluster centers from layer 4 for detailed interpretation: The left one with green box shows most nearest cell features to this cluster center are parts with legs shape, ranging from human legs to dogs legs. The right one with the red box shows most nearest cell features to this cluster center are parts of bird’s head or beetles, which share a dotted structure.
>
> Q4. "Experiments demonstrate that larger numbers of clusters yield stronger performance, suggesting that a coarse superpixel type clustering might be more promising than hoping to identify object parts across images."
>
> A4. Thank you for the suggestion. In Figure 2.a, we study the algorithm performance with respect to the number of the clusters, we observe a general trend that increasing the numbers for the clusters leads to improved performance. However, the performance plateaus at certain numbers.
>
> Using superpixels is an interesting direction to explore. In ConstellationNet, we design a modularized, recursive and general-purpose module to integrate to CNN, referring to Figure 2 (c,d) layer indices with constellation.
>
> Q5. "Visualization shows the discovered clusters are more appearance oriented than identifying parts (cf brown vs white on dog instead of recognising parts e.g. ears)."
>
> A5. We include a new visualization, Figure 3  (revised paper), to eliminate the confusion. In the visualization, we show the clusters are able to identify object parts, such as legs and head, beyond appearance oriented.
>
> Q6. "The clustering process itself lacks clarity. Cluster center fixed after training?"
>
> A6. Yes, the cluster centers are fixed after training to keep the parts that have been learned. Besides, we add a section in appendix A.8 to clarify the procedure of cell feature clustering.
>
> Q7. "Optimal number of clusters affected by the number of classes? Or similarity between classes"
>
> A7. Thank you for bringing this up. We include a new experiment in Table 4 (revised paper’s appendix). The result shows the optimal number of clusters are less affected by the number of clusters but more affected by the similarity between classes. Less number of clusters are needed for dataset with classes of high similarity. The highest accuracy occurs at 16 clusters (1-shot: 39.12% in ResNet-12). This aligns with our intuition, limited number of patterns exist in this dataset so that small number of clusters are enough to represent its part-based information.
>
> Please check Table 4 (revised paper’s appendix) for details.
>
> Q8. "Cluster receiving assignments for each batch?"
>
> A8. Yes. Although we don't reject assignments for each batch, the cluster features are dynamically weighted from the current batch, referring to Equation 5, 6 in the paper.

---

> > ### Author Response · Authors · 2020-11-20
> > **Response to reviewer 3 (Cont.)**
> >
> > Q9. "Batches constructed to optimize cluster learning?"
> >
> > A9. We randomly sample the batches from the dataset following standard SGD optimization in CNN training. Following (Sculley 2010), we do not add special design to construct the batches in order to optimize the cluster learning.
> >
> > (Sculley 2010) David Sculley. Web-scale k-means clustering. In Proceedings of the 19th international conference on World wide web, pp. 1177–1178, 2010.
> >
> > Q10. "expand on the role of count parameters."
> >
> > A10. Thank you for the suggestion. The counter $\mathbf{s}$ tracks the number of historical (soft) assignments to clusters. We use the counter to control the amount of adjustment (i.e. $\eta$) from each mini-batch: If there are already many assignments to the cluster center $\mathbf{v}_k$ now (i.e. large $s_k+\Delta s_k$), we update the cluster center with a small $\eta$ to avoid too much adjustment from the current mini-batch in order to stabilize the cluster center $\mathbf{v}_k$.
> >
> > Q11. "Are different clusters relevant to e.g. dog classes different from clusters for bird classes?"
> >
> > A11. Shown in the figure 3 (revised paper), the cluster might include parts from different classes. The leg cluster includes legs from human legs, unicycle wheels with human legs, dog legs.
> >
> >
> > Q12. "Why apply self-attention on a distance map ?"
> >
> > A12.  This is a good point. We apply self-attention on the distance map $\mathbf{D}$ because of two advantages: (1) the distance map is considered as abstracted code map after codebook learning (clustering), which is different from raw cell features. (2) when applying to the raw cell features $\mathbf{U}$,  self-attention would be limited to original feature dimensions, which leads to a quadratic computation w.r.t. feature dimension.
> >
> > For experiment details, see A5 response to Reviewer 4, and A1 response to Reviewer1. In short, applying self-attention on the distance map (1-shot: 57.03% in Conv-4) achieves better performance than directly applying it to the raw cell features (i.e. convolutional feature map) (1-shot: 55.92% in Conv-4).
> >
> >
> > Q13. "Why is cell relation a valid alternative for old school constellation models? What we hoped to learn using this strategy? Why this strategy can be viewed as a constellation type model."
> >
> > A13. Old school constellation models have the following pipeline: keypoint extraction-> codebook learning -> joint spatial relationship modeling.
> > The constellation module in ConstellationNet has the following pipeline: (dense) cell features  -> clustering ->  self-attention for the joint modeling of learned codes with positional encoding and their spatial relationship.
> >
> > In our ConstellationNet, after the cell feature clustering, each cell is represented by learned codes (distance vectors) + positional encoding, and the self-attention mechanism jointly captures the appearance and spatial configuration across the cells. Our model can be viewed as a constellation type model due to the same design principles: Codebook learning (analogous to) clustering; joint spatial relationship modeling (analogous to) self-attention on learned codes + positional encoding.
> >
> > Please see Section 4.4 in the updated version for more explanations and explained intuitions.
> >
> > Q14. "Few-shot learning formulation should be more clearly explained."
> >
> > A14. Thank you for pointing it out. We add an extended description of our few-shot learning framework in Appendix A.1 and clarify several key concepts including episodic training, meta-training/meta-test and the general ProtoNet model.

---

> > > ### Author Response · Authors · 2020-11-20
> > > **Response to reviewer 3 (Cont.)**
> > >
> > >
> > > Q15. "Relating the model and its components more closely to constellation models, and justification as to why the proposed strategy is a better implementation of constellations in deep learning framework than Simon et al. 2015"
> > >
> > > A15. Thank you for pointing it out.
> > >
> > > Our work is different from (Simon et al. 2015) in three aspects:
> > >
> > > 1. The algorithmic mechanisms behind (Simon et al. 2015) and ours are different: (Simon et al. 2015) implements a traditional Gaussian-based constellation module to model the spatial configuration and part selection on top of a fixed pre-trained CNN. However, in our ConstellationNet, our part representation and spatial configuration are modeled by cell feature clustering and self-attention based cell relation module, which is general-purpose, modularized and recursive.
> > > 2. In (Simon et al. 2015), the constellation module is optimized in an EM-like algorithm, which is separate from the CNN optimization. Our constellation modules are seamlessly integrated into the current CNNs and jointly optimized with them.
> > > 3. Our ConstellationNet uses the dense cell features from the CNN feature maps, which consider all positions from the images as potential parts and model their relation. However, (Simon et al. 2015) extracts sparse part representations (i.e. it uses at most one part proposal per channel and selects even less parts later), which may not fully utilize the rich information from the CNN feature maps.
> > >
> > > More discussions about the work with the constellation models are provided in section 4.4 (revised version). Please also see A2 to reviewer 1.
> > >
> > > Q16. "Providing clarifications regarding design decisions, experimental setting, and more intuition. In particular regarding the distance map and cluster centers. If possible, pay more attention to interpretability/observed behaviour of cluster centers in the experimental section."
> > >
> > > A16. We addressed design decisions, decision, experiment settings, intuition why our model yields good performance, design choice of distance map and cluster centers from A1 to A15 above.
> > >
> > > We include more interpretation and observation of cluster centers, two new added visualizations. (1) Visualization of the cell assignment and attention maps in Figure 4 (revised paper) (2) Visualization of features belonging to certain clusters for each layer in Conv-4 in Figure 3 (revised paper).
> > >
> > >
> > > Q17. "Experiments are missing important details. For example, it is not specified for experiments in Figure 2 and 3 which dataset and parameter configurations are used. In particular for figure 3, is the number of clusters set to 64? Are all cluster centers relevant to a given class?"
> > >
> > > A17. Thank you for bringing this up. For experiments in Figure 2 (original paper), we use the mini-ImageNet as the dataset with default parameter configuration: Conv-4, 64 clusters, 8-head attention and all layers are coupled with a constellation module. When we experiment with one hyper-parameter, other hyper-parameters remain as default. For experiments in Figure 3 (original paper), which is Figure 4 (revised paper), the number of clusters is set to 64. Cluster centers are not necessarily related to class, which is addressed with additional experiment in A7 above and Figure 3 (revised paper).
> > >
> > >
> > > Q18. "Examples from the same class provided in this figure look very similar in appearance. What happens when examples of the same class look different? Are the same cluster patterns observed?"
> > >
> > > A18. We have updated more diverse visualization to eliminate confusion. See Figure 4 Left (revised paper).
> > >
> > >
> > > Q19. "The multi-branch training strategy is not new and was suggested in TADAM, Oreshkin et al., NeurIPS, 2018."
> > >
> > > A19. The strategy used in TADAM is different from our multi-branch strategy. In TADAM, an auxiliary loss is co-trained to ease the difficulty to optimize both convolutional filters and the introduced TEN module. However, the backbone of TADAM is still the same as the standard prototypical network and only one embedding from the network is used for prediction. In our multi-branch training strategy, we modify prototypical networks into a one-stem-two-branch Y-shape architecture where prototypical training schemes and standard classification training schemes are optimized in each branch separately and jointly optimized in the stem network blocks. During testing, two separate embeddings from both branches are concatenated to make predictions. The motivation is that we found the two training schemes, standard classification and prototypical, can be a companion rather than a conflict when training in a multi-task fashion. So both the network architecture and motivation are different from TADAM.

---

> > > > ### Author Response · Authors · 2020-11-22
> > > > **Response to reviewer 3 (Cont.)**
> > > >
> > > > Q20. "Regarding ablation experiments, it would be interesting to see the influence of having a single module on the last layer (where levels of abstraction would be higher) vs modules at every layer."
> > > >
> > > > A20.  This is a good question. Thanks. Taking your advice, we experimented on the impact of having the constellation module at differ layers in a reversed order shown in Figure 2d (revised paper). We make the following observations: 1) having all four layers gives rise to the best performance (e.g. Conv-4 in 1-shot 57.03 %  improved from 50.62 %); 2) constellation module at layer 4 alone has already attained a good performance gain (e.g. Conv-4 in 1-shot 55.87 % improved from 50.62 %).
> > > >
> > > > Q21. "claims regarding ‘explicit modelling of parts’ should be revised. There is no explicit part discovery (nor a guarantee that object parts are indeed discovered), nor a clear, explicit modelling of their interactions. Maybe a more accurate characterisation would be that the approach integrates spatial information between image regions of similar appearance/texture. Similarly, it is not obvious that CNNs extract object parts."
> > > >
> > > >
> > > > A21. Although there is no guarantee that object parts are discovered, in our paper, 'explicit modeling of parts', in another term, is to design an explicit procedure that is able to model tokenized representations as expressive parts. In ConstellationNet, we use a cell feature clustering as the explicit procedure to obtain learned codes (distance vectors), together with the positional encoding as the the tokenized representations.
> > > >
> > > > Please see Figure 3 (revised paper), our modeled parts are not restricted to image regions of similar appearance/texture, but also can relate to regions related to some abstraction, such as legs (human legs, dog legs and other legs in Figure 3).

---

### Official Review · AnonReviewer1 · 2020-10-29
**Connection to constellation models is interesting**

**Rating:** 6
**Confidence:** 5

**Review:**

This paper proposes ConstellationNet for few-shot learning, which is inspired by constellation models.  The constellation models firstly model local appearance of an image by visual codebooks, and then the local appearance and spatial configuration of keypoints are learned by generative models. The proposed ConstellationNet is a neural network that combines codebook learning and self-attention models, not generative models. The way of using spatial information is different from the constellation models. The experimental results show that the ConstellationNet outperforms state-of-the-art methods in the few-shot learning problem. Overall, the paper is well written, and the connection to the constellation model is interesting.

Pros
+  The historical explanation of spatial parts-based models in computer vision researches is extensive and interesting. The writing of this paper is mostly clear.

+ The mini-batch soft k-means is suitable to conduct codebook learning in the process of CNN training with SGD.

+ The proposed ConstellationNet shows higher accuracies than state-of-the-art few-shot learning methods on standard mini-ImageNet/CIFAR-FS/FC100 datasets.

+ Ablation study shows the performance improvements are not because of increasing model parameters. The ConstellationNet adds additional layers to the backbone network, and increasing model size improves performance is obvious. To answer this concern, the authors compared with the case when the model size increased differently, and the constellation model outperformed that case.

+ Visualization of codebooks shows similar parts are assigned to the same cluster.

Cons
- My main concern is that if the clustering of cell features are really needed. It is known that the convolutional features correspond to (implicit) object parts.
Though the similar points are activated in the same cluster in Fig.3, original convolutional channels would be consistently activated in the same semantic points.
The ablation study leaks the case when the codebook is not used. The recognition accuracies of ConstellationNet should be compared with the case when the codebook learning is not used. Also, 1x1 convolution can be used to aggregate different channel information and reduce the dimensions, instead of codebook learning.  This case also should be compaerd.

- If the proposed method can be called a constellation model is somewhat questionable. The self-attention only learns feature channel weights for computing similarity of different cells. It does not explicitly learn the spatial configuration of local cells of an object category, unlike the constellation models.

- Visualization of attention is lacking. What spatial relations among the cell features are learned should be shown.

---

> ### Author Response · Authors · 2020-11-20
> **We thank reviewer 1 for the valuable comments. Please see our answers to the questions below.**
>
> We thank reviewer 1 for the valuable comments. Please see our reply below.
>
> Q1. "My main concern is if the clustering of cell features are really needed. It is known that the convolutional features correspond to (implicit) object parts. Though the similar points are activated in the same cluster in Fig.3, original convolutional channels would be consistently activated in the same semantic points. The ablation study leaks the case when the codebook is not used. The recognition accuracies of ConstellationNet should be compared with the case when the codebook learning is not used. Also, 1x1 convolution can be used to aggregate different channel information and reduce the dimensions, instead of codebook learning. This case also should be compared."
>
> A1. Thank you for your suggestion. We conduct the experiment with the cases you mentioned. As shown in Table 3 (revised paper), we have following findings:
>
> - Applying self-attention on the distance map (1-shot: 57.03% in Conv-4) achieves better performance than directly applying it to the original cell features (i.e. convolutional feature map) (1-shot: 55.92% in Conv-4) .
>
> - In addition, we also tried to replace the cell feature clustering module with a 1x1 convolution layer (output dimension is equal to the number of clusters)  (1-shot: 55.46% in Conv-4). It is worse than our results (1-shot: 57.03% in Conv-4) as well. We observe that the 1x1 convolution layer is less expressive than the cell feature clustering module, making it difficult to extract enough context information during cell relation modeling.
>
> Please also see response A5 to reviewer 4.
>
> Q2. "If the proposed method can be called a constellation model is somewhat questionable. The self-attention only learns feature channel weights for computing similarity of different cells. It does not explicitly learn the spatial configuration of local cells of an object category, unlike the constellation models."
>
> A2. Thank you for the comment. This is a good point to bring up. The spatial configuration has been modeled in our constellation module. The self-attention is performed on the clustering map with positional encoding for each cell. Motivated by Transformer (Vaswani et al., 2017), our constellation module takes advantage of self-attention by modeling both tokenized image representation (distance map/code) and spatial relationships (positional encoding) within the self-attention unit. We have added positional encoding illustration into Figure 1 (revised paper).
>
> We have revised Figure 1 and added a new section 4.4 to describe more about the intuition and explanation for our method in the revised paper.
>
>
> Q3. "Visualization of attention is lacking. What spatial relations among the cell features are learned should be shown."
>
> A3. Thanks for the suggestion. We have added a new visualization to show spatial relations among cell features learned by the self-attention module in Figure 4 Right (revised paper).  We visualize attention maps of one query feature (at the location of the red point in the left part) with all key features. It shows that all the attention heads together can extract the features of the whole object.

---

### Official Review · AnonReviewer4 · 2020-10-30
**I lean toward acceptance. But I have few questions.**

**Rating:** 6
**Confidence:** 5

**Review:**

Summary

The paper proposes a constellation model that performs feature clustering and encoding dense part representations. The constellation module is placed after convolutional blocks. The module clusters cell features and calculates distance map between each cluster centroids and cell feature. The self-attention mechanism is applied on the distance map and concatenated to the original feature map to complement the feature representation. The resulting feature representation contains part representations. The few-shot experiments on the mini-Imagenet, CIFAR-FS, and FC100 datasets show the effectiveness of the proposed method.


Rating

Overall, I lean toward accepting the paper. The paper suggests multiple technical components to improve few-shot performance. Among the contributions, a constellation module which is the main idea of the paper plays a major role in the performance gain. The performance is competitive and ablation studies show the importance of constellation module in the performance gain. There are a few questions while reading the paper but not critical.

Pros
-      The proposed module is plugged into the backbone network. The authors verified the effectiveness of the approach on both 4-conv networks and resnet-12 networks.
-      The paper showed several techniques to improve the performance including the constellation module, multi-branch structure, multi-head attention and feature augmentation.
-      The proposed approach performs favourably against the competing methods.
-      Ablation studies show that most of performance gain comes from the constellation module, which is the main idea of the paper while the other technical contributions give additional performance gain.

Cons (minor issues)
-      The combination of multiple components requires more hyper-parameters to tune. (number of clusters, number of heads in attention layer, constellation module locations)
-      The figure2.(c) shows the constellation module analysis graph. The graph shows more constellation modules results in better performance. However, it is not reported how the model performs when the constellation modules are located in higher layers only. (for example, the module located at only after layer 4.)

Questions
-      Although the constellation module is using a clustering algorithm, there is no clustering loss term. I conjecture the network using the constellation module to converge much slower than without the module even though the module does not bring a significant amount of parameter increment. How does the module affect the training time?
-      Euclidean distance is used for clustering and distance map calculation (equation 4). Is there any reason to choose Euclidean distance rather than the cosine similarity?
-      In the attention calculation, a dot product is used on the distance map (equation 8). What is the intuition to apply self-attention on the distance map ‘D’ rather than the cell features ‘U’? Distances are the difference between each cluster centroids and cell features. Isn’t cell feature map is more suitable for attention mechanism?

---

> ### Author Response · Authors · 2020-11-20
> **We thank reviewer 4 for the valuable comments. Please see our answers to the questions below.**
>
> We thank reviewer 4 for the constructive comments. Please see our answers to the questions below.
>
> Q1. "The combination of multiple components requires more hyper-parameters to tune. (number of clusters, number of heads in attention layer, constellation module locations)"
>
> A1. This is a good question. Indeed, in Figure 2 (original paper), we study the algorithm performance with respect to specified values for various hyper-parameters including the number of the clusters, the number of heads in the attention layer, and the number of layers with constellations. From the figures, we observe a general trend that increasing the numbers for the clusters, heads, constellation layers leads to improved performance. However, the performance plateaus at certain numbers. There are indeed a couple of more hyper-parameters for our constellation module but the complexity in hyper-parameter tuning is reasonable based on the reported study.
>
>
> Q2. "The Figure 2 (c) shows the constellation module analysis graph. The graph shows more constellation modules results in better performance. However, it is not reported how the model performs when the constellation modules are located in higher layers only (for example, the module located at only after layer 4)."
>
> A2. This is a good question. Thank you. Taking your advice, we experimented on the impact of having the constellation module at differ layers in a reversed order shown in Figure 2d (revised paper). We make the following observations: 1) having all four layers gives rise to the best performance (e.g. Conv-4 in 1-shot 57.03%  improved from 50.62%); 2) constellation module at layer 4 alone has already attained a good performance gain (e.g. Conv-4 in 1-shot 55.87% improved from 50.62%).
>
>
> Q3.  "Although the constellation module is using a clustering algorithm, there is no clustering loss term. I conjecture the network using the constellation module to converge much slower than without the module even though the module does not bring a significant amount of parameter increment. How does the module affect the training time?"
>
> A3. Thank you for bringing this issue up. The training time of Conv-4 baseline is 26.2 mins, where Conv-4 with Constellation takes 96 mins. The training time of ResNet-12 baseline is 1.8 hrs, where ResNet-12 with Constellation takes 3.2 hrs. The experiment on Conv-4 model is run on a single RTX 2080 Ti GPU, and the experiment on ResNet-12 is run on dual RTX 2080 Ti GPUs. The additional training time of our constellation model brings a large performance gain ( Conv-4 in 1-shot 57.03 %  improved from 50.62 % ).  There is indeed no clustering loss term in our algorithm. The clustering algorithm is fully integrated into the CNN network pipeline. This has become a more viable direction since capsule networks (CapsNets) (Sabour et al., 2017).
>
>
> Q4.  "Euclidean distance is used for clustering and distance map calculation (equation 4). Is there any reason to choose Euclidean distance rather than the cosine similarity?"
>
> A4. Euclidean distance and cosine similarity are equivalent up to an affine transformation (i.e. $ d( \mathbf{u}_i, \mathbf{v}_k) = ||\mathbf{u}_i-\mathbf{v}_k||^2_2 = 2-2\mathbf{u}_i^\top\mathbf{v}_k = 2-2\text{cos}(\mathbf{u}_i, \mathbf{v}_k)$) in our implementation, since we applied L2 normalization to the cell feature $\mathbf{u}_i$ and the cluster center $\mathbf{v}_k$ (as mentioned in Section 4.1).
>
> Q5. "In the attention calculation, a dot product is used on the distance map (Eq. 8). What is the intuition to apply self-attention on the distance map 'D' rather than the cell features 'U'? Distances are the difference between each cluster centroids and cell features. Isn't a cell feature map more suitable for an attention mechanism?"
>
> A5.  We apply self-attention on the distance map $\mathbf{D}$ because of two advantages: (1) the distance map is considered as an abstraction for the parts (like in the classical bag of words models), which is different from raw cell features. (2) when applying to the raw cell features $\mathbf{U}$,  self-attention would be limited to original feature dimensions, which leads to a quadratic computation w.r.t. feature dimension.
>
> In addition, we conduct an additional ablation study on the effectiveness of the cell feature clustering module in the Table 3 (revised paper). We have following findings: Applying self-attention on the distance map (1-shot: 57.03% in Conv-4) achieves better performance than directly applying it to the raw cell features (i.e. convolutional feature map) (1-shot: 55.92% in Conv-4) .
>
> For additional experiment details, see A1 response to Reviewer1.
>
> Please see Section 4.4 (updated version) for more discussions and explained intuitions.

---

### Author Response · Authors · 2020-11-23
**General response to all the reviewers.**

We thank all the reviewers for their detailed comments and suggestions!

The main contribution of our work is a constellation module/block that performs cell-wise clustering, followed by self-attention on the clustering distance map + positional encoding for each cell. This separates our work from previous attempts.

Here, we summarize the update in the paper and address several frequently asked questions among reviewers:

(1) We mainly update/include the following materials in the paper:
- Update Figure 1 (Illustration of our Constellation pipeline):
  - Details of cell feature clustering are re-drawn to show how distance map (learned cell code map) is constructed from cell features and cluster centers;
  - Positional encoding is included to illustrate that the self-attention module captures the spatial relationship among parts, implying the spatial configuration.
- Update Section 2 (Related work):
   - Include citations for part-based models on fine-grained image classification and object detection (Simon & Rodner, 2015; Peng et al., 2017; Zhu et al., 2017; Ge et al., 2019; Qi et al., 2019).
   - Include the discussion and comparison between our ConstellationNet and Neural Activation Constellations (Simon & Rodner, 2015).
 - Include Section 4.4 Explanation (Why clustering and self-attention (clustering map+positional encoding)):
   - We justify our design choice and main contribution.
 - Update Table 1 (Comparison to prior work on mini-imagenet):
      - AM3 (Xing et al., 2019), AM3 + TRAML (Li et al., 2020), Negative Margin (Liu et al., 2020) are included in the comparison. $\dagger$ shows Xing et al., 2019, Li et al., 2020 results use extra cross-modal information.
 - Update Table 3 (Effectiveness of modules):
      - Row 4 and row 5 are added to the table.
      - Our Constellation applies self-attention on the distance map (row 6: 57.03% in Conv-4 1-shot) achieves better performance than directly applying it to the raw cell features (i.e. convolutional feature map) (row 4: 55.92% in Conv-4 1-shot), and a 1x1 convolution layer baseline (output dimension is equal to the number of clusters) (row 5: 55.46% in Conv-4 1-shot).
 - Update Figure 2 (Ablation study)
      - Figure 2 (d) is added to the Figure. In Figure 2 (c,d) together show results from both clustering algorithms applied to both early features and high-level features yields better performance than applying to early or high-level features alone.
 - Include Figure 3 (Visualization of cluster centers)
      - We show the clusters are able to identify object parts, such as legs and head, beyond appearance oriented.
 - Update Figure 4 Left (Visualization of the cells assignment)
      - Figures from bird, dog categories are updated to show more diverse results.
 - Include Figure 4 Right (Visualization of attention maps)
      - We choose one query feature at the center of the object and attend it to all key features.
      - It is observed that some parts are identified such as head, legs, buttock and body. A merged attention map by overlaying all 8 attention maps indicates that all the attention heads together can extract the features of the whole object, which would be used for final classification.

(2) Why does ConstellationNet yield good performance?

 - Despite the effectiveness of the CNN models in standard supervised image classification, their transfer abilities are limited when training data for novel classes is scarce in few-shot learning tasks. We take the inspiration from the traditional constellation models, and come up with a realization to gain greater modeling capability.

- Analogous to classical constellation models (Fergus et al., 2003; Felzenszwalb & Huttenlocher, 2005) to extract parts with their joint spatial relationships, we perform codebook (dictionary) learning by cell clustering followed by self-attention on the code/clustering map and positional encoding for each cell. Seamlessly integrating constellation models with convolution operations improves the generalization in representation over the baseline CNNs for few-shot learning.

---

> ### Author Response · Authors · 2020-11-23
> **General response to all the reviewers (Cont.)**
>
>
> (3) Why clustering and self-attention (clustering map+positional encoding)?
>
>  - The main contribution of our work is a constellation module/block that performs cell-wise clustering, followed by self-attention on the clustering distance map + positional encoding for each cell. This separates our work from previous attempts, e.g.  non-local block work (Wang et al., 2018) in which long-range non-linear averaging is performed on the convolution features (no clustering, nor positional encoding for the spatial configuration). The main properties of our constellation block include:
>     1. Cell based dense representation as opposed to the sparse part representation in (Weber et al., 2000) to make the cells recursively modeled in the self-attention unit in a modularized and general-purpose way.
>     2. Clustering to generate the cell code after clustering  (codebook learning) that attains abstraction and is not dependent on the CNN feature dimensions.
>     3. Positional encoding (as in (Vaswani et al., 2017)) for cells to capture the spatial configurations.
>     4. Tokenized representation as expressive parts (code/clustering distance map + positional encoding) for the cells.
>     5. Self-attention to jointly model the cell code and positional encoding to capture the joint relationships between the parts together with their spatial configurations.
>  - More details can be found from from Section 4.4
>
> (4) Compare to other works
>
> - Traditional constellation methods
>
>     - Old school constellation models have the following pipeline: keypoint extraction -> codebook learning -> joint spatial relationship modeling. The constellation module in ConstellationNet has the following pipeline: (dense) cell features -> clustering -> self-attention for the joint modeling of learned codes with positional encoding and their spatial relationship.
>
>     - In our ConstellationNet, after the cell feature clustering, each cell is represented by learned codes (distance vectors) + positional encoding, and the self-attention mechanism jointly captures the appearance and spatial configuration across the cells. Our model can be viewed as a constellation type model due to the same design principles: Codebook learning (analogous to) clustering; joint spatial relationship modeling (analogous to) self-attention on learned codes + positional encoding.
>
> - Non-local blocks
>
>     - There is a large difference between our work and that of Wang et al. 2018. In non-local neural networks (Wang et al. 2018), not only is clustering/codebook learning absent, there is no positional encoding that encodes spatial configuration involved in the method. In this sense, the non-local neural network work (Wang et al., 2018) differs from our in multiple aspects including:
>       1. Motivation (theirs is to capture long-distance dependency beyond local operations vs. ours to model the parts and their spatial relations);
>       2. Implementation (theirs is to have self-attention on raw flattened CNN features vs. ours to have self-attention on clustering map + positional encoding);
>       3. Modules (no clustering and no positional/spatial encoding in Wang et al. 2018);
>       4. Domain (video classification vs. static images for few-shot learning).

---

### Decision · Program_Chairs · 2021-01-07
**Final Decision**

**Decision:**

Accept (Poster)

**Comment:**

This paper proposes a meta-learning method that learns structured features based on constellation modules. Exploiting object parts and their relationships is a promising direction for few-shot learning as AnonReviewer3 described. The effectiveness of the proposed method is demonstrated with experiments using standard benchmark, and ablation study.